# ON THE PERILS OF CASCADING ROBUST CLASSIFIERS

**Ravi Mangal**[*], **Zifan Wang**[*], **Chi Zhang**[*]
Electrical and Computer Engineering
Carnegie Mellon University
Pittsburgh, PA 15213
`{rmangal, zifanw, chiz5}@andrew.cmu.edu`

**Klas Leino**
School of Computer Science
Carnegie Mellon University
Pittsburgh, PA 15213
`kleino@cs.cmu.edu`

**Corina Păsăreanu**
Carnegie Mellon University
and NASA Ames
Moffett Field, CA 94043
`pcorina@andrew.cmu.edu`

**Matt Fredrikson**
School of Computer Science
Carnegie Mellon University
Pittsburgh, PA 15213
`mfredrik@cmu.edu`

## ABSTRACT

Ensembling certifiably robust neural networks is a promising approach for improving the *certified robust accuracy* of neural models. Black-box ensembles that assume only query-access to the constituent models (and their robustness certifiers) during prediction are particularly attractive due to their modular structure. Cascading ensembles are a popular instance of black-box ensembles that appear to improve certified robust accuracies in practice. However, we show that the robustness certifier used by a cascading ensemble is unsound. That is, when a cascading ensemble is certified as locally robust at an input $x$ (with respect to $\epsilon$), there can be inputs $x'$ in the $\epsilon$-ball centered at $x$, such that the cascade's prediction at $x'$ is different from $x$ and thus the ensemble is not locally robust. Our theoretical findings are accompanied by empirical results that further demonstrate this unsoundness. We present *cascade attack* (CasA), an adversarial attack against cascading ensembles, and show that: (1) there exists an adversarial input for up to 88% of the samples where the ensemble claims to be certifiably robust and accurate; and (2) the accuracy of a cascading ensemble under our attack is as low as 11% when it claims to be certifiably robust and accurate on 97% of the test set. Our work reveals a critical pitfall of cascading certifiably robust models by showing that the seemingly beneficial strategy of cascading can actually hurt the robustness of the resulting ensemble. Our code is available at https://github.com/TristaChi/ensembleKW.

## 1 INTRODUCTION

Local robustness has emerged as an important requirement of classifier models. It ensures that models are not susceptible to misclassifications caused by small perturbations to correctly classified inputs. A lack of robustness can be exploited by not only malicious actors (in the form of adversarial examples (Szegedy et al., 2014)) but can also lead to incorrect behavior in the presence of natural noise (Gilmer et al., 2019). However, ensuring local robustness of neural network classifiers has turned out to be a hard challenge. Although neural networks can achieve state-of-the-art classification accuracies on a variety of important tasks, neural classifiers with comparable certified robust accuracies[1] (CRA, Def. 2.2) remain elusive, even when trained in a robustness-aware manner (Madry et al., 2018; Wong & Kolter, 2018; Cohen et al., 2019; Leino et al., 2021). In light of the limitations of robustness-aware training, ensembling certifiably robust neural classifiers has been shown to be a promising approach for improving certified robust accuracies (Wong et al., 2018; Yang et al., 2022). An ensemble combines the outputs of multiple base classifiers to make a prediction, and is a well-known mechanism for improving classification accuracy when one only has access to weak learners (Dietterich, 2000; Bauer & Kohavi, 1999).

Ensembles designed to improve CRA take one of two forms. *White-box ensembles* (Yang et al., 2022; Zhang et al., 2019; Liu et al., 2020) assume white-box access to the constituent models. They

---

[*]Equal Contribution
[1]Percentage of inputs where the classifier is accurate and certified as locally robust.

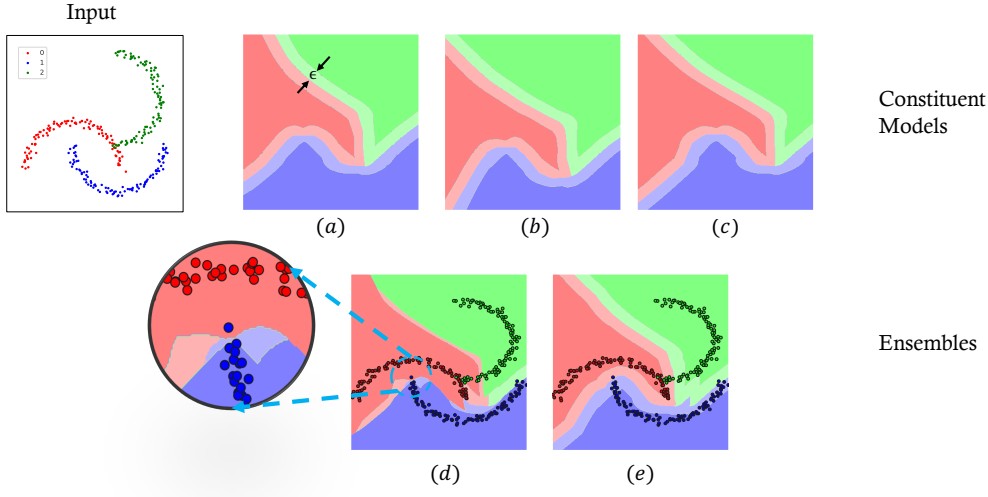

Figure 1: Visualizing classification results of 2D points for constituent models (a-c) and the corresponding Cascading Ensemble (d, Def. 2.7) and Uniform Voting Ensemble (e, Def. 5.3). Regions with colors correspond to predictions (0: red, 1: blue, 2: green) made by the underlying model (or ensemble). Darker colors indicate that the accompanying robustness certification of the underlying model (or ensemble) returns 1 and lighter colors are for cases when the certification returns 0. All points receiving 1 for certifications (darker regions) are at least $\epsilon$-away from the other classes in (a)-(c), i.e. *certification is sound* (Def. 2.3). This property is violated in (d), e.g. points from dark red regions are not $\epsilon$-away from the blue region in the zoomed-in view on the left, but preserved in (e). Namely, voting ensembles are *soundness-preserving* (Def. 2.6) while cascading ensembles are not.

calculate new logits by averaging the corresponding logits of the constituent classifiers. For local robustness certification, they treat the ensemble as a single, large model and then use off-the-shelf techniques (Cohen et al., 2019; Weng et al., 2018; Wong & Kolter, 2018; Zhang et al., 2018) for certification. *Black-box ensembles* (Wong et al., 2018; Blum et al., 2022), on the other hand, assume only query-access to the constituent classifiers during prediction, and are, therefore, agnostic to their internal details. They re-use the prediction and certification outcomes of the constituent models to calculate the ensemble's prediction and certificate. Their black-box nature lends them modularity and permits any combination of constituent classifiers, irrespective of their individual certification mechanism, so we focus our efforts on them in this paper.

*Cascading ensembles* (Wong et al., 2018; Blum et al., 2022) are a particularly popular instance of black-box ensembles that appear to improve `CRA` in practice. They evaluate the constituent classifiers (and their certifiers) in a fixed sequence. The ensemble's prediction is the output of the first constituent classifier in the sequence that is certified locally robust, defaulting to the last classifier's output if no model can be certified. Importantly, the cascading ensemble is itself certified locally robust only when at least one of the constituent classifiers is certified locally robust.

**Our contributions.** We show in this paper that the local robustness certification mechanism used by cascading ensembles is unsound even when the certifiers used by each of the constituent classifiers are sound (Theorem 2.8). In other words, when a cascading ensemble is certified as locally robust at an input $x$, there can, in fact, be inputs $x'$ in the $\epsilon$-ball centered at $x$, such that the cascade's prediction at $x'$ is different from $x$. Figure 1 demonstrates this visually on a toy dataset. The cascading ensemble can have points that are less than $\epsilon$ away from the decision boundary, yet the ensemble is certified locally robust at such points (Figure 1(d)). As a consequence of our result, use of a cascading ensemble in any scenario requiring local robustness guarantees is unsafe and existing empirical results that report the `CRA` of cascading ensembles are not valid.

Guided by our theoretical construction, we propose *cascade attack* (CasA, Algorithm 3.1), an adversarial attack against cascading ensembles, and conduct an empirical evaluation with the cascading ensembles trained by Wong et al. (2018) for MNIST and CIFAR-10 datasets. With CasA, we show that: (1) there exists an adversarial example for up to 88% of the samples where the ensemble claims

to be certifiably robust and accurate; (2) the empirical robust accuracy of a cascading ensemble is as low as 11% while it claims to be certifiably robust and accurate on 97% of the test set; and (3) viewing all experiments as a whole, the empirical robust accuracy of a cascading ensemble is almost always lower than even the CRA of the single best model in the ensemble. Namely, a cascading ensemble is often less robust. Our results conclusively demonstrate that the unsoundness of the cascading ensemble certification mechanism can be exploited in practice, and cause the ensemble to perform markedly worse than the single best constituent model.

We also present an alternate ensembling mechanism based on weighted voting that, like cascading ensembles, assumes only query-access to the constituent classifiers but comes with a provably sound local robustness certification procedure (Section 5). We show through a thought experiment that it is possible for a voting ensemble to improve upon the CRA of its constituent models (Section 5.2), and observe that the key ingredient for the success of voting ensembles is a suitable balance between diversity and similarity of their constituents. We leave the design of training algorithms that balance the diversity and similarity of the constituent classifiers as future work.

## 2 CASCADING ENSEMBLES

In this section, we introduce our notation and required definitions. We then show the local robustness certification procedure used by cascading ensembles is unsound.

### 2.1 CERTIFIABLE CLASSIFIERS AND ENSEMBLERS

We begin with our notation and necessary definitions. Suppose a neural network $f : \mathbb{R}^d \to \mathbb{R}^m$ takes an input and outputs the probability of $m$ different classes. The subscript $x_j$ denotes the $j$-th element of a vector $x$. When discussing multiple networks, we differentiate them with a superscript, e.g. $f^{(1)}, f^{(2)}$. Throughout the paper we use the upper-case letter $F$ to denote the prediction of $f$ such that $F(x) = \text{argmax}_{j \in \mathcal{Y}} \{f_j(x)\}$ where $f_j$ is the logit for class $j$ and $\mathcal{Y} = [m]$.[2] The prediction $F(x)$ is considered $\epsilon$-locally robust at $x$ if all neighbors within an $\epsilon$-ball centered at $x$ receive the same predictions, which is formally stated in Def. 2.1.

**Definition 2.1** ($\epsilon$-Local Robustness). A network, $F$, is $\epsilon$-locally robust at $x$ w.r.t to norm, $|| \cdot ||$, if $\forall x' . ||x' - x|| \leq \epsilon \implies F(x') = F(x)$.

Though local robustness certification of ReLU networks is NP-Complete (Katz et al., 2017), due to its importance, the problem has been receiving increasing attention from the community. Proposed certification methods rely on a variety of algorithmic approaches like solving corresponding linear (Jordan et al., 2019) or semi-definite programs (Raghunathan et al., 2018), interval propagations (Gowal et al., 2018; Lee et al., 2020; Zhang et al., 2018), abstract interpretation (Singh et al., 2019a), geometric projections (Fromherz et al., 2021), dual networks (Wong & Kolter, 2018), or Lipschitz approximations (Leino et al., 2021; Weng et al., 2018).

If a certification method is provided for a network $F$, we use $\tilde{F}^\epsilon : \mathbb{R}^d \to \mathcal{Y} \times \{0, 1\}$ to denote a *certifiable neural classifier* that returns a prediction according to $F$ and the outcome of the certification method applied to $F$ with respect to robustness radius $\epsilon$. We use $\tilde{F}^\epsilon_{\text{label}}(x)$ to refer to the prediction and $\tilde{F}^\epsilon_{\text{cert}}(x)$ to refer to the certification outcome. If $\tilde{F}^\epsilon_{\text{cert}}(x) = 0$, the accompanying robustness certification is unable to certify $F$ (i.e., $\tilde{F}_{\text{label}}$) as $\epsilon$-locally robust at $x$. When $\epsilon$ is clear from the context, we directly write $\tilde{F}$. One popular metric to evaluate the performance of any $\tilde{F}$ is Certified Robust Accuracy (CRA).

**Definition 2.2** (Certified Robust Accuracy). The certified robust accuracy (CRA) of a certifiable classifier $\tilde{F} \in \mathbb{R}^d \to \mathcal{Y} \times \{0, 1\}$ on a given dataset $S_k \subseteq \mathbb{R}^d \times \mathcal{Y}$ with $k$ samples is given by $\text{CRA}(\tilde{F}, S_k) := 1/k \sum_{(x_i, y_i) \in S_k} \mathbb{1}[\tilde{F}(x_i) = (y_i, 1)]$.

For $\tilde{F}$ and its CRA to be useful in practice without providing false robustness guarantees, it must be *sound* to begin with (Def. 2.3).

**Definition 2.3** (Certification Soundness). A certifiable classifier, $\tilde{F} : \mathbb{R}^d \to \mathcal{Y} \times \{0, 1\}$, is sound if $\forall x \in \mathbb{R}^d . \tilde{F}_{\text{cert}}(x) = 1 \implies \tilde{F}_{\text{label}}$ is $\epsilon$-locally robust at $x$.

---

[2] $[m] := \{0, 1, ..., m-1\}$

Notice that if there exist $\epsilon$-close inputs $x,x'$ where $\tilde{F}(x) = (y,1)$ and $\tilde{F}(x') = (y',0)$, where $y \neq y'$, then it still means that $\tilde{F}$ is not sound. We define an *ensembler* (Def. 2.4) as a function that combines multiple certifiable classifiers into a single certifiable classifier (i.e., an *ensemble*).

**Definition 2.4** (Ensembler)**.** Let $\tilde{\mathbb{F}} := \mathbb{R}^d \to \mathcal{Y} \times \{0,1\}$ represent the set of all certifiable classifiers. An ensembler $\mathcal{E} : \tilde{\mathbb{F}}^N \to \tilde{\mathbb{F}}$ is a function over $N$ certifiable classifiers that returns a certifiable classifier.

A *query-access* ensembler formalizes our notion of a black-box ensemble.

**Definition 2.5** (Query-Access Ensembler)**.** Let $\mathbb{G} := (\mathcal{Y} \times \{0,1\})^N \to \mathcal{Y} \times \{0,1\}$. $\mathcal{E}$ is a query-access ensembler if, $\forall \tilde{F}^{(0)}, \tilde{F}^{(1)}, ..., \tilde{F}^{(N-1)} \in \tilde{\mathbb{F}} . \exists G \in \mathbb{G} . \forall x \in \mathbb{R}^d.$

$$\mathcal{E}\Big(\tilde{F}^{(0)}, \tilde{F}^{(1)}, ..., \tilde{F}^{(N-1)}\Big)(x) = G\Big(\tilde{F}^{(0)}(x), \tilde{F}^{(1)}(x), ..., \tilde{F}^{(N-1)}(x)\Big)$$

Def. 2.5 says that if $\mathcal{E}$ is *query-access* its output $\tilde{F}$ can always be re-written as a function over the outputs of the certifiable classifiers $\tilde{F}^{(0)}, \tilde{F}^{(1)}, ..., \tilde{F}^{(N-1)}$. Put differently, $\tilde{F}$ only has black-box or query-access to classifiers $\tilde{F}^{(0)}, \tilde{F}^{(1)}, ..., \tilde{F}^{(N-1)}$.

Finally, a *soundness-preserving* ensembler (Def. 2.6) ensures that if the constituent certifiable classifiers are sound (as defined in Def. 2.3), the ensemble output by the ensembler is also sound.

**Definition 2.6** (Soundness-Preserving Ensembler)**.** An ensembler $\mathcal{E}$ is soundness-preserving if, $\forall \tilde{F}^{(0)}, \tilde{F}^{(1)}, ..., \tilde{F}^{(N-1)} \in \tilde{\mathbb{F}}, \tilde{F}^{(0)}, \tilde{F}^{(1)}, ..., \tilde{F}^{(N-1)}$ are sound $\implies \mathcal{E}(\tilde{F}^{(0)}, \tilde{F}^{(1)}, ..., \tilde{F}^{(N-1)})$ is sound.

## 2.2 Cascading Ensembler Is Not Soundness-Preserving

Cascading ensembles (Wong et al., 2018; Blum et al., 2022) are a popular instance of black-box ensembles that appear to be practically effective in improving certified robust accuracies. However, we show that cascading ensembles are not sound.

We define a cascading ensemble to be the output of a *cascading ensembler* (Def. 2.7). A cascading ensemble evaluates its constituent certifiable classifiers in a fixed sequence. For a given input $x$, the ensemble either returns the prediction and certification outcomes of the first constituent classifier $\tilde{F}^{(j)}$ such that $\tilde{F}^{(j)}_{\text{cert}} = 1$ or of the last constituent classifier in case none of the constituent classifiers can be certified. Clearly, cascading ensemblers are query-access (formal proof in Appendix A).

**Definition 2.7** (Cascading Ensembler)**.** Let $\tilde{F}^{(0)}, \tilde{F}^{(1)}, ..., \tilde{F}^{(N-1)}$ be $N$ certifiable classifiers. A cascading ensembler $\mathcal{E}_C : \tilde{\mathbb{F}}^N \to \tilde{\mathbb{F}}$ is defined as follows

$$\mathcal{E}_C\Big(\tilde{F}^{(0)}, \tilde{F}^{(1)}, ..., \tilde{F}^{(N-1)}\Big)(x) := \begin{cases} \tilde{F}^{(j)}(x) & \text{if } \exists j \leq N-1 . \mathsf{c}(j) = 1 \\ \tilde{F}^{(N-1)}(x) & \text{otherwise} \end{cases}$$

where $\mathsf{c}(j) := 1$ if $(\tilde{F}^{(j)}_{\text{cert}}(x) = 1)$ and $(\forall i < j, \tilde{F}^{(i)}_{\text{cert}}(x) = 0)$, and 0 otherwise.

Theorem 2.8 shows that cascading ensemblers are not soundness-preserving, and so a cascading ensemble can be unsound. We show this by means of a counterexample.[3]

**Theorem 2.8.** *The cascading ensembler $\mathcal{E}_C$ is not soundness-preserving.*

*Proof.* We can re-write the theorem statement as, $\exists \tilde{F}^{(0)}, \tilde{F}^{(1)}, ..., \tilde{F}^{(N-1)} \in \tilde{\mathbb{F}}$ such that for $\tilde{F} := \mathcal{E}_C(\tilde{F}^{(0)}, \tilde{F}^{(1)}, ..., \tilde{F}^{(N-1)}), \exists x \in \mathbb{R}^d, \tilde{F}_{\text{cert}}(x) = 1 \not\Longrightarrow \tilde{F}_{\text{label}}$ is $\epsilon$-locally robust at $x$.

We prove by constructing the following counterexample. Consider a cascading ensemble $\tilde{F}$ constituted of certifiable classifiers $\tilde{F}^{(0)}$ and $\tilde{F}^{(1)}$. $\tilde{F}^{(0)}$ and $\tilde{F}^{(1)}$ are such there exists an $x$ where

$$\Big(\tilde{F}^{(0)}_{\text{cert}}(x) = 0\Big) \wedge \Big(\tilde{F}^{(1)}(x) = (y,1)\Big) \tag{1}$$

Using Def. 2.7, it is true that $\tilde{F}(x) = (y,1)$. Without violating (1), we can have another point $x'$ such that,

---

[3]The use of a cascade of certification methods for a single classifier as in (Gowal et al., 2018; Singh et al., 2019b) is orthogonal and sound.

---

**Algorithm 3.1:** Cascade Attack (CasA)

---

**Inputs:** Ensemble $\tilde{F} \in \tilde{\mathbb{F}}$, constituent models $\tilde{F}^{(0)}, ..., \tilde{F}^{(N-1)} \in \tilde{\mathbb{F}}$, input $x \in \mathbb{R}^d$, attack bound $\epsilon \in \mathbb{R}$, and distance metric $\ell_p$

**Output:** An attack input $x' \in \mathbb{R}^d$

1   `Attack`$(\tilde{F}, \tilde{F}^{(0)}, ..., \tilde{F}^{(N-1)}, x, \epsilon, \ell_p)$**:**

2     $y := \tilde{F}_{\mathsf{label}}(x)$

3     $\mathtt{idxs} := \{i \mid i \in [N] \wedge \tilde{F}_{\mathsf{label}}^{(i)}(x) \neq y\} \cup \{N-1\}$

4     **foreach** $i \in \mathtt{idxs}$ **do**

5       **if** $i = N-1$ **then**

6         $x^* := x + \underset{\delta \in \mathbb{B}_p(0,\epsilon)}{\mathrm{argmax}} \left( \mathcal{L}^{ce}(\mathsf{one\text{-}hot}(\tilde{F}_{\mathsf{label}}^{(i)}(x+\delta)), \mathsf{one\text{-}hot}(y)) + \sum_{k<i}(1 - \tilde{F}_{\mathsf{cert}}^{(k)}(x+\delta)) \right)^4$

7       **else**

8         $x^* := x + \underset{\delta \in \mathbb{B}_p(0,\epsilon)}{\mathrm{argmax}} \left( \tilde{F}_{\mathsf{cert}}^{(i)}(x+\delta) + \sum_{k<i}(1 - \tilde{F}_{\mathsf{cert}}^{(k)}(x+\delta)) \right)^4$

9       **if** $\tilde{F}_{label}(x^*) \neq y$ **then**

10         **return** $x^*$

11     **return** $x$

---

$$(||x - x'|| \leq \epsilon) \wedge (\tilde{F}^{(0)}(x') = (y',1)) \wedge (\tilde{F}^{(1)}(x') = (y,0)) \wedge (y' \neq y) \qquad (2)$$

Using Def. 2.7, it is true that $\tilde{F}(x') = (y', 1)$. Thus, for two points $x, x'$ constructed as above, we show that $\exists x, x'$, s.t. $||x' - x|| \leq \epsilon$, $\tilde{F}_{\mathsf{cert}}(x) = 1 \implies \tilde{F}_{\mathsf{label}}(x') = \tilde{F}_{\mathsf{label}}(x)$, which violates the condition of local robustness (Def. 2.1). $\qquad \square$

The counterexample constructed in Thm. 2.8 is not just hypothetical, but something that materializes on real models (see Figure 1 for a toy example and Section 4 for our empirical evaluation).

## 3   ATTACKING CASCADING ENSEMBLES

Section 2.2 shows that a cascading ensemble does not provide a robustness guarantee. We further show here how one can attack the cascading ensemble and find an adversarial example within the $\epsilon$-ball centered at the input $x$.

**Overview of Attack.** Algorithm 3.1 describes the attack algorithm, *cascade attack* (CasA), inspired by the proof of Theorem 2.8. Given an input $x$, the goal of the algorithm is to find an input $x'$ in the $\epsilon$-ball centered at $x$ such that the predictions of the cascade at $x$ and $x'$ are different. The inputs to the algorithm are an ensemble $\tilde{F}$, its constituent classifiers $\tilde{F}^{(0)}, ..., \tilde{F}^{(N-1)}$, the input $x$ to be attacked, and the attack distance bound $\epsilon$ as well as distance metric $\ell_p$. The algorithm either returns a successful adversarial input $x'$ such that $||x - x'||_p \leq \epsilon$ and $\tilde{F}_{\mathsf{label}}(x) \neq \tilde{F}_{\mathsf{label}}(x')$ or it returns the original input $x$ if no adversarial input was found. We use the following notations: $\mathcal{L}^{ce}$ stands for cross-entropy loss, $\mathsf{one\text{-}hot}$ is the one-hot encoding function, and $\mathbb{B}_p(0, \epsilon)$ is the $\ell_p$-ball of radius $\epsilon$ centered at 0.

**Preparing Targets.** CasA gets the label $y$ predicted by the ensemble $\tilde{F}$ at input $x$ (line 2) to select the constituent models it may attack. The attacker is only interested in a constituent model (by remembering its index $i$) if it predicts a label other than $y$ at $x$ or it is the last one (line 3). We are not interested in attacking a model $\tilde{F}^{(j)}$ that predicts $y$ at $x$ because such an effort is bound to fail. $\tilde{F}^{(j)}$ is still sound even though the ensemble is not; therefore, no point $x'$ assigned a label other than $y$ by $\tilde{F}^{(j)}$ is such that it is both less than $\epsilon$-away from $x$ and $\tilde{F}^{(j)}$ is also certifiably robust at $x'$ (the second condition is necessary for $\tilde{F}^{(j)}$ to be used for prediction at $x'$ by the ensemble). However, the last model $\tilde{F}^{(N-1)}$ is an exception and always remembered, i.e. $\mathtt{idxs}$ always includes $N-1$. The reason is that, given an input $x'$, if all models $\tilde{F}^{(i)}; i < N-1$ fail to be certifiably robust at $x'$, the ensemble uses $\tilde{F}^{(N-1)}$ for prediction at $x'$ irrespective of whether $\tilde{F}^{(N-1)}$ is itself certifiably robust at $x'$ or not.

---

[4]In our implementation, we use a surrogate version of this objective (see Section 3).

**Attacker's Steps.** For each model index in `idxs`, we try to find an adversarial example (lines 4-10). An attacker stops as soon as they find a valid adversarial example (lines 9-10). Lines 6 and 8 describe the objective an attacker minimizes to find the adversarial examples. If index $i \neq N-1$, the attacker optimizes $\delta$ such that, at input $x+\delta$, the model $\tilde{F}^{(i)}$ is certified robust whereas all other models $\tilde{F}^{(k)}; k < i$ are not certified robust. This ensures that model $\tilde{F}^{(i)}$ is used for prediction at input $x+\delta$ as it is certifiably robust at $x$. If index $i = N-1$, we still require that all models $\tilde{F}^{(k)}; k < i$ are not certified robust at $x+\delta$. But instead of requiring that $\tilde{F}^{(i)}$ is certified robust at $x+\delta$, we only require that the predicted label at $x+\delta$ be different from $y$. We solve the optimization problems on lines 6 and 8 using projected gradient descent (PGD) (Madry et al., 2018).

**Surrogate Objectives.** For cases when the certification procedure, i.e. $\tilde{F}^{(i)}_{\mathsf{cert}}(x+\delta)$, is not differentiable or too expensive to run multiple times, we provide the following cheap surrogate replacements. The intuition underlying the surrogate versions is that, given a model, the distance to the decision boundary from an input is correlated with the margin between the top logit scores of the model at that input. To use the surrogate objectives for the attack, we need to assume access to the logit scores of the models. For the problem $\mathrm{argmax}_{\delta \in \mathbb{B}_p(0,\epsilon)} \tilde{F}^{(i)}_{\mathsf{cert}}(x+\delta)$, we try to increase the logit score associated with the desired prediction as much as possible. Then, a surrogate version of the problem is as follows (where $\tilde{F}^{(i)}_{\mathsf{logit}}$ represents the logit scores produced by model $\tilde{F}^{(i)}$):

$$\underset{\delta \in \mathbb{B}_p(0,\epsilon)}{\mathrm{argmax}} -\mathcal{L}^{ce}(\tilde{F}^{(i)}_{\mathsf{logit}}(x+\delta), \mathsf{one\text{-}hot}(\tilde{F}^{(i)}_{\mathsf{label}}(x))) \tag{3}$$

For the problem $\mathrm{argmax}_{\delta \in \mathbb{B}_p(0,\epsilon)} \sum_{k<i}(1 - \tilde{F}^{(k)}_{\mathsf{cert}}(x+\delta))$, we want the input $x+\delta$ to be as close as possible to the decision boundaries for each of the models by $\tilde{F}^{(k)}, k < i$ so that the robustness certifications will fail. The specific predictions $\tilde{F}^{(k)}_{\mathsf{label}}(x+\delta)$ of these models do not matter. Towards that end, we aim to make the margin between the logit scores of any model $\tilde{F}^{(k)}$ be as small as possible. This leads to the following surrogate problem (where $\mathsf{unif}$ is a discrete uniform distribution):

$$\underset{\delta \in \mathbb{B}_p(0,\epsilon)}{\mathrm{argmax}} -\sum_{k<i}\mathcal{L}^{ce}(\tilde{F}^{(k)}_{\mathsf{logit}}(x+\delta), \mathsf{unif}) \tag{4}$$

## 4 EMPIRICAL EVALUATION

The goal of our empirical evaluation is to demonstrate the extent to which the unsoundness of the cascading ensembles manifests in practice and can be exploited by an adversary, i.e. CasA. For our measurements, we use the $\ell_\infty$ and $\ell_2$ robust cascading ensembles constructed by Wong et al. (2018) for MNIST (LeCun et al., 1998) and CIFAR-10 (Krizhevsky, 2009) datasets. The constituent classifiers in these cascades use a local robustness certification mechanism based on dual networks (Wong et al., 2018). Each cascade includes between 2-7 certifiable classifiers with the same architecture (except for the $\ell_\infty$ robust, CIFAR-10 Resnet cascades that include only a single constituent model, and are hence not considered in our evaluation). The training code and all the constituent models in the ensembles are made available by Wong et al. (2018) in a public repository (Wong & Kolter).

We report the certified robust accuracy (CRA) and standard accuracy (Acc) for the cascading ensemble as well as the single best constituent model in the ensemble. While the certifier for a single model is sound, the ensemble certifier is unsound and the reported ensemble CRA is an over-estimate. We therefore measure the empirical robustness of the ensemble under CasA. Certifying with dual networks (Wong et al., 2018) is differentiable but extremely expensive. To run the attack more efficiently, we use the surrogate replacements in Section 3 and take 100 steps using PGD (other hyper-parameters to follow in Appendix B) to empirically minimize the objectives. After the attack, we report the false positive rate (FPR), i.e. % of test inputs for which an adversarial example is found within the $\epsilon$-ball, and the empirical robust accuracy (ERA), i.e. % of test set where the cascade is empirically robust (i.e., our attack failed). All our experiments were run on a pair of NVIDIA TITAN RTX GPUs with 24 GB of RAM each, and a 4.2GHz Intel Core i7-7700K with 64 GB of RAM.

Table 1 shows the results for $\ell_\infty$ robustness (top) and $\ell_2$ robustness (bottom). Each row in the table represents a specific combination of dataset (MNIST or CIFAR-10), architecture (Small or Large convolutional networks), and $\epsilon$ value used for local robustness certification. The structure of the table is the same as Tables 2 and 4 in (Wong et al., 2018), except we add the columns reporting FPR and ERA.

**Summary of Results.** We see from Table 1 that, irrespective of the dataset, model, $\epsilon$ value, or $\ell_p$ metric under consideration, our attack can find false positives, with false positive rates (FPR) as

Table 1: Results on models pre-trained by Wong et al. (2018) for $\ell_\infty$ (top) and $\ell_2$ (bottom) robustness. CRA: % of test set where model is certified robust and accurate. Acc: % of test set where model is accurate. FPR: among all test inputs where cascade is certified robust and accurate, % of inputs for which an adversarial example is found within the $\epsilon$-ball using our ensemble attack (i.e., false positive rate). ERA: % of test set where the cascading is empirically robust (i.e., our attack failed) and accurate (ERA of a single model is always greater or equal to its CRA because of *soundness* and therefore not included). The unsoundness of cascade certification is shown by the high false positive rates (FPR).

| $\ell_\infty$ Dataset | Model | $\epsilon$ | Single Model | | Cascading Ensemble | | | |
|---|---|---|---|---|---|---|---|---|
| | | | CRA(%) | Acc(%) | unsound CRA(%) | FPR(%) | Acc(%) | ERA(%) |
| MNIST | Small, Exact | 0.1 | 95.54 | 98.96 | 96.33 | 88.71 | 96.62 | 11.17 |
| MNIST | Small | 0.1 | 94.94 | 98.79 | 96.07 | 81.93 | 96.24 | 17.51 |
| MNIST | Large | 0.1 | 95.55 | 98.81 | 96.27 | 86.37 | 96.42 | 13.27 |
| MNIST | Small | 0.3 | 56.21 | 85.23 | 65.41 | 88.87 | 65.80 | 7.67 |
| MNIST | Large | 0.3 | 58.02 | 88.84 | 65.50 | 85.27 | 65.50 | 9.65 |
| CIFAR10 | Small | 2/255 | 46.43 | 60.86 | 56.65 | 11.51 | 56.65 | 50.13 |
| CIFAR10 | Large | 2/255 | 52.65 | 67.70 | 64.87 | 10.47 | 65.13 | 58.15 |
| CIFAR10 | Small | 8/255 | 20.58 | 27.60 | 28.32 | 16.00 | 28.32 | 23.79 |
| CIFAR10 | Large | 8/255 | 16.04 | 19.01 | 20.83 | 17.67 | 20.83 | 17.15 |

| $\ell_2$ Dataset | Model | $\epsilon$ | Single Model | | Cascading | | | |
|---|---|---|---|---|---|---|---|---|
| | | | CRA(%) | Acc(%) | unsound CRA(%) | FPR(%) | Acc(%) | ERA(%) |
| MNIST | Small, Exact | 1.58 | 43.52 | 88.14 | 75.58 | 44.72 | 80.43 | 43.46 |
| MNIST | Small | 1.58 | 43.34 | 87.73 | 74.66 | 40.93 | 79.07 | 45.73 |
| MNIST | Large | 1.58 | 43.96 | 88.39 | 74.50 | 51.95 | 74.99 | 35.81 |
| CIFAR10 | Small | 36/255 | 46.05 | 54.39 | 49.89 | 3.61 | 51.37 | 49.27 |
| CIFAR10 | Large | 36/255 | 50.26 | 60.14 | 58.72 | 2.70 | 58.76 | 57.17 |
| CIFAR10 | Resnet | 36/255 | 51.65 | 60.70 | 58.65 | 3.41 | 58.69 | 56.68 |

Table 2: Run time and peak memory usage of CasA. Results are reported on one Titan RTX.

| Dataset | Model | # of Models | $\epsilon$ | Objective | unsound CRA(%) | ERA(%) | Batch Size | Mem./Batch (MB) | Time/Batch (Min) |
|---|---|---|---|---|---|---|---|---|---|
| MNIST | Small | 7 | 0.3 ($\ell_\infty$) | dual networks | 65.41 | 4.93 | 32 | 12565 | 4.34 |
| MNIST | Small | 7 | 0.3 ($\ell_\infty$) | our surrogates | 65.41 | 7.67 | 32 | 4548 | 0.06 |
| CIFAR10 | Small | 5 | 2/255 ($\ell_\infty$) | dual networks | 28.32 | 23.79 | 32 | 10124 | 2.43 |
| CIFAR10 | Small | 5 | 2/255 ($\ell_\infty$) | our surrogates | 28.32 | 22.92 | 32 | 4898 | 0.06 |

high as 88.87%. In other words, there always exist test inputs where the ensemble is accurate and declares itself to be certified robust, but our attack is able to find an adversarial example. This result demonstrates that the unsoundness of the cascading ensemble certification mechanism is not just a problem in theory but it can be exploited by adversaries in practice. More strikingly, the empirical robust accuracy (ERA) of the ensemble is often significantly lower than the certified robust accuracy (CRA) of the best constituent model. Since the ERA of a model is an upper-bound of its CRA, the actual CRA of the ensemble can be no larger than the reported ERA. This result shows that the use of a cascading ensemble can actually hurt the robustness instead of improving it. Note that there are small differences between the Acc and CRA numbers reported in Table 1 and those in (Wong et al., 2018). Though we use the evaluation code and pre-trained models made available by Wong et al. (2018), the hardware and PyTorch versions we use in our experiments are different.

**Attack Efficiency.** In Table 2, we compare the attack results of CasA using the original objectives, i.e. dual networks (Wong et al., 2018), and using surrogate replacements. Because the ensemble on MNIST contains more constituent models, it uses more memory with dual networks compared to CIFAR10. Our report of run time and memory usage shows that using surrogate replacements allows us to run attacks with larger batch size, less memory and time to reach the same level of performance.

## 5    A QUERY-ACCESS, SOUNDNESS-PRESERVING ENSEMBLER

We present a query-access, soundness-preserving ensembler based on weighted voting in this section. Voting is a natural choice for the design of a query-access ensembler but ensuring that the ensembler is soundness-preserving can be subtle. Section 5.1 defines our ensembler and proves that it is soundness-preserving. In Section 5.2, we present a thought experiment demonstrating that it is possible for a voting

ensemble to significantly improve upon the CRA of its constituent models. Appendix D describes our algorithm for learning the weights to be used in weighted voting, and Appendix E presents initial empirical results with the voting ensemble. Our results show that improving the CRA via a voting ensemble can be difficult on realistic datasets since it requires the ensemble to demonstrate a suitable balance between diversity and similarity of its constituents, but we believe that this is a fruitful direction for future research.

## 5.1 WEIGHTED VOTING ENSEMBLE

Voting ensembles run a vote amongst their constituent classifiers to make a prediction. In the simplest case, each constituent has a single vote that gets assigned to their predicted label. The label with the maximum number of votes is chosen as the ensemble's prediction. More generally, weighted voting allows a single constituent to be allocated more than one vote. The decimal number of votes allocated to each constituent is referred to as its *weight*. For simplicity, we assume that weights of the constituents in an ensemble are normalized so that they sum up to 1. We use weighted voting to not only choose the ensemble's prediction but to also decide its certification outcome. The interaction between voting and certification is subtle and needs careful design to ensure that the certification procedure is sound.

**Extra Notations.** Let $v_x^w(j,c)$ denote the total number of votes allocated to certifiable classifiers, $\tilde{F}^{(i)}$, in the ensemble that output $(j,c)$. More formally, for an input $x$, label $j \in \mathcal{Y}$, certification outcome $c \in \{0,1\}$, weight $w \in [0,1]^N$, and a set of constituent certifiable classifiers, $\tilde{F}^{(0)}, ..., \tilde{F}^{(N-1)}$, let $v_x^w(j,c) := \sum_{i=0}^{N-1} w_i * \mathbb{1}[\tilde{F}^{(i)}(x) = (j,c)]$. We find it useful to use $v_x^w(*,c)$ to denote the number of votes for any class with certificate, $c$; i.e., $v_x^w(*,c) = \sum_{j \in \mathcal{Y}} v_x^w(j,x)$. Likewise, we will use $v_x^w(j)$ to denote the number of votes for class $j$ regardless of certificate; i.e., $v_x^w(j) = v_x^w(j,0) + v_x^w(j,1)$.

**Definition 5.1** (Weighted Voting Ensembler). Let $\tilde{F}^{(1)}, ..., \tilde{F}^{(N)}$ be $N$ certifiable classifiers. A weighted voting ensembler, $\mathcal{E}_V^w : \tilde{\mathbb{F}}^N \to \tilde{\mathbb{F}}$ is defined as follows

$$\tilde{F}(x) := \mathcal{E}_V^w(\tilde{F}^{(0)}, \tilde{F}^{(1)}, ..., \tilde{F}^{(N-1)})(x) := \left( \tilde{F}_{\mathsf{label}}(x), \tilde{F}_{\mathsf{cert}}(x) \right), \quad \tilde{F}_{\mathsf{label}}(x) := \underset{j}{\mathrm{argmax}} \left\{ v_x^w(j) \right\}^5$$

$$\text{where } \tilde{F}_{\mathsf{cert}}(x) := \begin{cases} 1 & \text{if } \forall j \neq \tilde{F}_{\mathsf{label}}(x) \ . \ v_x^w(\tilde{F}_{\mathsf{label}}(x),1) > v_x^w(*,0) + v_x^w(j,1) \\ 0 & \text{otherwise} \end{cases}$$

The prediction of the weighted voting ensemble is the label receiving the maximum number of votes regardless of the certificate. However, for the certification outcome, the ensemble has to consider the certificates of the constituent models. The ensemble should be certified robust only if its prediction outcome, i.e., the label receiving the maximum number of votes (regardless of the certificate), can be guaranteed to not change in an $\epsilon$-ball. The condition under which $\tilde{F}_{\mathsf{cert}}(x) = 1$ ensures this is the case, and allows us to prove that weighted voting ensemblers are soundness-preserving (Theorem 5.2). A key observation underlying the condition is that only constituent classifiers that are not certified robust at the current input can change their predicted label in the $\epsilon$-ball, and, in the worst case, transfer all their votes ($v_x^w(*,0)$) to the label with the second highest number of votes at $x$. We believe that our proof of soundness-preservation is of independent interest. We also note that weighted-voting ensemblers are query-access (formal proofs in Appendix A).

**Theorem 5.2.** *The weighted voting ensembler $\mathcal{E}_V^w$ is soundness-preserving.*

**Definition 5.3** (Uniform Voting Ensembler). Let $\tilde{F}^{(0)}, ..., \tilde{F}^{(N-1)}$ be $N$ certifiable classifiers. The uniform voting ensembler, $\mathcal{E}_U : \tilde{\mathbb{F}}^N \to \tilde{\mathbb{F}}$ is a weighted voting ensembler that assigns equal weights to each classifier, i.e. $\mathcal{E}_U = \mathcal{E}_V^w$ where $\forall i \in \{0, ..., N-1\}. w_i = 1/N$.

## 5.2 EFFECTIVENESS OF VOTING: A THOUGHT EXPERIMENT

Voting ensembles require the constituents to strike the right balance between diversity and similarity to be effective. In other words, while the constituents should be accurate and robust in different regions of the input space (diversity), these regions should also have some overlap (similarity). We conduct a thought experiment using a simple hypothetical example (Example. 5.4) where such a balance is struck. The existence of this example provides evidence and hope that voting ensembles can improve the CRA. We present the example informally here. The detailed, rigorous argument is in Appendix A.

---

[5] In case of a tie, we assume that the label corresponding to the logit with the lowest index is returned.

**Example 5.4.** *Assume that we have a uniform voting ensemble $\tilde{F}$ with three constituent classifiers $\tilde{F}^{(0)}$, $\tilde{F}^{(1)}$, and $\tilde{F}^{(2)}$. Assume that on a given dataset $S_k$ with 100 samples, each of the constituent classifiers has* CRA *equal to 0.5. Let's say that the samples in $S_k$ are ordered such that $\tilde{F}^{(0)}$ is accurate and robust on the first 50 samples (i.e., samples 0-49), $\tilde{F}^{(1)}$ is accurate and robust on samples 25-74, and $\tilde{F}^{(2)}$ on samples 0-24 and 50-74. Then, for each of the first 75 samples, two out of three constituents in the ensemble are accurate and robust. Therefore, by Def. 5.1, the ensemble $\tilde{F}$ is accurate and robust on samples 0-74, and has* CRA *equal to 0.75.*

## 6 RELATED WORK

Ensembling is a well-known approach for improving the accuracy of models as long as the constituent models are suitably diverse (Dietterich, 2000). In recent years, with the growing focus on *robust accuracy* as a metric of model quality, a number of ensembling techniques have been proposed for improving this metric. Depending on whether an ensemble is query-access or not (i.e., does not or does require access to the internal details of the constituent models for prediction and certification), it can be classified as a *white-box* or a *black-box* ensemble. The modularity of black-box ensembles is attractive as the constituent classifiers can each be from a different model family (i.e., neural networks, decision trees, support vector machines, etc.) and each use a different mechanism for robustness certification. The constituents of white-box ensembles, on the other hand, tend to be from the same model family but this provides the benefit of tuning the ensembling strategy to the model family being used.

**White-box ensembles.** Several works (Yang et al., 2022; Zhang et al., 2019; Liu et al., 2020) present certifiable ensembles where the ensemble logits are calculated by averaging the corresponding logits of the constituent classifiers. Needing access to the logits of the constituent classifiers, and not just their predictions, is one aspect that makes these ensembles white-box. More importantly, the approaches used by these ensembles for local robustness certification are also in violation of our definition of query-access ensembles (Def. 2.5). For instance, randomized smoothing (Cohen et al., 2019) is used in (Yang et al., 2022; Liu et al., 2020) to certify the ensemble, which requires evaluating the constituent models on a large number of inputs for each prediction, and not just one input. Other approaches (Zhang et al., 2019) use interval bound propagation (IBP) (Gowal et al., 2018; Zhang et al., 2018) to certify the ensemble. Calculating the interval bounds requires access to the architecture and weights of each of the constituent models, violating the requirements of a query-access ensemble. A number of white-box ensembling techniques (Pang et al., 2019; Yang et al., 2020; Kariyappa & Qureshi, 2019; Sen et al., 2020; Zhang et al., 2022) only aim to improve *empirical* robust accuracy, i.e., these ensembles do not provide a robustness certificate. As before, the ensemble logits are calculated by averaging the corresponding logits of the constituent models. These approaches differ from each other in the training interventions used to promote diversity in the constituent models.

**Black-box ensembles.** Cascading ensembles (Wong et al., 2018; Blum et al., 2022) are the most popular example of certifiably robust black-box ensembles. While Wong et al. (2018) empirically evaluate their cascading ensemble, the results of Blum et al. (2022) are purely theoretical. However, as we show in this work, the certification mechanism used by cascading ensembles is unsound. Devvrit et al. (2020); Sen et al. (2020) present a black-box voting ensemble but, unlike our voting ensemble, their ensemble does not provide robustness certificates. Nevertheless, they are able to show improvements in the empirical robust accuracy with the voting ensemble.

## 7 CONCLUSION

In this paper, we showed that the local robustness certification mechanism used by cascading ensembles is unsound. As a consequence, existing empirical results that report the certified robust accuracies (CRA) of cascading ensembles are not valid. Guided by our theoretical results, we designed an attack algorithm against cascading ensembles and demonstrated that their unsoundness can be easily exploited in practice. In fact, the performance of the ensembles is markedly worse than their single best constituent model. Finally, we presented an alternate black-box ensembling mechanism based on weighted voting that we prove to be sound, and, via a thought experiment, showed that voting ensembles can significantly improve the CRA if the constituent models have the right balance between diversity and similarity.

## ETHICS STATEMENT

Our work sheds light on existing vulnerabilities in state-of-the-art certifiably robust neural classifiers. The presented attacks can be used by malicious entities to adversarially attack deployed cascading ensembles of certifiably robust models. However, by putting this knowledge out in the public domain and making practitioners aware of the existence of the problem, we hope that precautions can be taken to protect existing systems. Moreover, it highlights the need to harden future systems against such attacks.

## REPRODUCIBILITY STATEMENT

To examine our theoretical results, the proof of Theorem 2.8 directly follows the body of the theorem in Section 2.2 while the proof of Theorem 5.2 is delayed to Appendix A and F, together with the proofs of other theorems that only appear in the appendix, i.e. Theorem A.1, A.2 (Appendix A) and Theorem F.2 (Appendix F). All the datasets used in our work are publicly available with links in their corresponding reference. Our experimental code is uploaded in the supplementary material (and also at `https://github.com/TristaChi/ensembleKW`) with a detailed README file and weights of models to reproduce the results in Table 1, 2, 5, 6, 7, and 8. Moreover, hyper-parameters used in these table are also documented in Appendix B and D. The hardware information used in all experiments is reported in Section 4.

## ACKNOWLEDGEMENTS

We would like to thank the reviewers for their comments which helped us improve this article. The work described in this paper has been supported by the Software Engineering Institute under its FFRDC Contract No. FA8702-15-D-0002 with the U.S. Department of Defense, DARPA and the Air Force Research Laboratory under agreement number FA8750-15-2-0277, as well as DARPA GARD Contract HR00112020006.

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

# A   PROOFS

**Theorem 5.2.** *The weighted voting ensembler $\mathcal{E}_V^w$ is soundness-preserving.*

*Proof.* Let $\tilde{F}^{(0)}, ..., \tilde{F}^{(N-1)}$ be $N$ certifiable classifiers, which we assume are sound.   Let $\tilde{F} := \mathcal{E}_V^w(\tilde{F}^{(0)}, ..., \tilde{F}^{(N-1)})$; i.e., $\tilde{F}$ is given by Definition 5.1.

Assume for the sake of contradiction $\exists x, x'$ s.t. $||x - x'|| \le \epsilon$, $\tilde{F}(x) = (j_1, 1)$, and $\tilde{F}_{\mathsf{label}}(x') = j_2$ where $j_2 \ne j_1$.

Since $\tilde{F}(x) = (j_1, 1)$, by Definition 5.1., $\forall j \ne j_1$, $v_x^w(j_1, 1) > v_x^w(*, 0) + v_x^w(j, 1)$, and thus, in particular, Equation 5 holds.

$$v_x^w(j_1, 1) > v_x^w(*, 0) + v_x^w(j_2, 1) \tag{5}$$

Consider the votes on $x'$. The models that contribute to $v_x^w(j_1, 1)$ are all locally robust[6] at $x$, so each of these models must output the label $j_1$ on $x'$, which is at distance no greater than $\epsilon$ from $x$; thus Equation 6 holds.

$$v_{x'}^w(j_1) \ge v_x^w(j_1, 1) \tag{6}$$

Conversely, only points that are non-robust at $x$ can change labels on $x'$, thus we obtain Equation 7.

$$v_{x'}^w(j_2) \le v_x^w(*, 0) + v_x^w(j_2, 1) \tag{7}$$

Putting things together we have

$$\begin{aligned} v_{x'}^w(j_1) &\ge v_x^w(j_1, 1) & \text{by (6)} \\ &> v_x^w(*, 0) + v_x^w(j_2, 1) & \text{by (5)} \\ &\ge v_{x'}^w(j_2) & \text{by (7)} \end{aligned}$$

Thus, since $v_{x'}^w(j_1) > v_{x'}^w(j_2)$, $\tilde{F}_{\mathsf{label}}(x')$ cannot be $j_2$ ⨑.        $\square$

**Example 5.4.** *We want to show that, $\exists \tilde{F}^{(0)}, \tilde{F}^{(1)}, ..., \tilde{F}^{(N-1)} \in \tilde{\mathbb{F}}, w \in [0, 1]^N$ such that for $\tilde{F} := \mathcal{E}_C^w(\tilde{F}^{(0)}, \tilde{F}^{(1)}, ..., \tilde{F}^{(N-1)})$,*

$$\exists S_k \subseteq \mathbb{R}^d \times \mathcal{Y}. \forall i \in \{0, ..., N-1\}. \; CRA(\tilde{F}, S_k) > CRA(\tilde{F}^{(i)}, S_k)$$

*Consider a weighted voting ensemble $\tilde{F}$ constituted of certifiable classifiers $\tilde{F}^{(0)}, \tilde{F}^{(1)},$ and $\tilde{F}^{(2)}$ with weights $w = (\frac{1}{3}, \frac{1}{3}, \frac{1}{3})$, i.e., $\tilde{F}$ is a uniform voting ensemble.*

*Suppose $k = 100$, i.e., $S_k$ is a dataset with 100 samples.   Moreover, lets say that $CRA(\tilde{F}^{(0)}, S_k) = CRA(\tilde{F}^{(1)}, S_k) = CRA(\tilde{F}^{(2)}, S_k) = 0.5$. Also, suppose that the samples in $S_k$ are arranged in a fixed sequence $S_k = (x_0, y_0), ..., (x_{k-1}, y_{k-1})$ such that,*

$$\forall i \in [0, 49]. \; \tilde{F}^{(0)}(x_i) = (y_i, 1) \tag{8}$$

$$\forall i \in [25, 74]. \; \tilde{F}^{(1)}(x_i) = (y_i, 1) \tag{9}$$

$$\forall i \in [0, 24] \cup [50, 74]. \; \tilde{F}^{(2)}(x_i) = (y_i, 1) \tag{10}$$

*where $[i, j]$ is the set of integers from $i$ to $j$, $i$ and $j$ included. 8, 9, and 10 are consistent with the fact that certified robust accuracy of each model is 0.5.*

*From 8, 9, 10, and Definition 5.1,*

$$\forall i \in [0, 74]. \; \tilde{F}_{\mathit{label}}(x_i) = \underset{j}{\mathrm{argmax}} \left\{ v_{x_i}^w(j) \right\} = y_i \tag{11}$$

$$\forall i \in [0, 74]. \; \forall j \ne y_i . \; v_{x_i}^w(y_i, 1) > v_{x_i}^w(*, 0) + v_{x_i}^w(j, 1) \tag{12}$$

*From 12 and Definition 5.1,*

$$\forall i \in [0, 74]. \tilde{F}_{\mathit{cert}}(x_i) = 1 \tag{13}$$

*From 11, 13, and Definition 2.2,*

$$CRA(\tilde{F}, S_k) = 0.75 \tag{14}$$

**Theorem A.1.** *The cascading ensemble $\mathcal{E}_C$ is query-access.*

---

[6]The models are known to be locally robust because they are *sound* and their output matches $(*, 1)$.

Table 3: Hyper-parameters Used for CasA in Table 1

| Dataset | Norm | $\epsilon$ | Normalization | Max Steps | Step Size |
|---------|------|------------|---------------|-----------|-----------|
| MNIST | $\ell_\infty$ | 0.1 | [0,1] | 100 | 0.004 |
| MNIST | $\ell_\infty$ | 0.3 | [0,1] | 100 | 0.012 |
| MNIST | $\ell_2$ | 1.58 | [0,1] | 100 | 0.03 |
| CIFAR10 | $\ell_\infty$ | 2/255 | $\mu$=[0.485, 0.456, 0.406], $\sigma$=0.225 | 100 | 0.0003 |
| CIFAR10 | $\ell_\infty$ | 8/255 | $\mu$=[0.485, 0.456, 0.406], $\sigma$=0.225 | 100 | 0.00124 |
| CIFAR10 | $\ell_2$ | 36/255 | $\mu$=[0.485, 0.456, 0.406], $\sigma$=0.225 | 100 | 0.0003 |

*Proof.* Let $\tilde{F} := \mathcal{E}_C(\tilde{F}^{(0)}, \tilde{F}^{(1)}, ..., \tilde{F}^{(N-1)})$. Let $g^{(0)}, g^{(1)}, ..., g^{(N-1)} \in \mathcal{Y} \times \{0,1\}$. We use $g_2^{(j)}$ to refer to the second element in the pair $g^{(j)}$. Define $G$ as follows,

$$G(g^{(0)}, g^{(1)}, ..., g^{(N-1)}) := \begin{cases} g^{(j)} & \text{if } \exists j \leq N-1. \ \mathtt{c}(j)=1 \\ g^{(N-1)} & \text{otherwise} \end{cases}$$

where $\mathtt{c}(j) := 1$ if $(g_2^{(j)}=1) \wedge (\forall i < j, g_2^{(i)}=0)$ and 0 otherwise.

Then, by Def. 2.7, $\tilde{F}(x) = G(\tilde{F}^{(0)}(x), \tilde{F}^{(1)}(x), ..., \tilde{F}^{(N-1)}(x))$. Then, by Def. 2.5, cascading ensembler is query-access. $\square$

**Theorem A.2.** *The weighted voting ensemble $\mathcal{E}_V^w$ is query-access.*

*Proof.* Let $\tilde{F} := \mathcal{E}_V^w(\tilde{F}^{(0)}, \tilde{F}^{(1)}, ..., \tilde{F}^{(N-1)})$. Let $g^{(0)}, g^{(1)}, ..., g^{(N-1)} \in \mathcal{Y} \times \{0,1\}$. We use $\bar{g}$ to refer to the set $\{g^{(0)}, g^{(1)}, ..., g^{(N-1)}\}$. Define $G$ as follows,

$$G(g^{(0)}, g^{(1)}, ..., g^{(N-1)}) := \big(G_1(g^{(0)}, g^{(1)}, ..., g^{(N-1)}), G_2(g^{(0)}, g^{(1)}, ..., g^{(N-1)})\big)$$

where

$$\hat{j} = G_1(g^{(0)}, g^{(1)}, ..., g^{(N-1)}) := \operatorname*{argmax}_j \Big\{ v_{\bar{g}}^w(j,*) \Big\},$$

$$G_2(g^{(0)}, g^{(1)}, ..., g^{(N-1)}) := \begin{cases} 1 & \text{if } \forall j \neq \hat{j} \ . \ v_{\bar{g}}^w(\hat{j},1) > v_{\bar{g}}^w(*,0) + v_{\bar{g}}^w(j,1) \\ 0 & \text{otherwise} \end{cases}$$

and

$$v_{\bar{g}}^w(j,c) = \sum_{i=0}^{N-1} w_i * \mathbb{1}\Big[ g^{(i)} = (j,c) \Big]$$

Then, by Def. 5.1, $\tilde{F}(x) = G(\tilde{F}^{(0)}(x), \tilde{F}^{(1)}(x), ..., \tilde{F}^{(N-1)}(x))$. Then, by Def. 2.5, weighted voting ensembler is query-access. $\square$

## B  HYPER-PARAMETERS OF TABLE 1

In Table 3, we report hyper-parameters used to run CasA to reach the statistics reported in Table 1. Notice that if a normalization is '$\mu$=[0.485, 0.456, 0.406], $\sigma$=0.225', we divide the $\epsilon$ and step size by $\sigma$ during the experiment. We use SGD as the optimizer for all experiments.

## C  ATTACKING NON-SEQUENTIALLY TRAINED CASCADING ENSEMBLES

Wong et al. (2018) train cascading ensembles in a sequential manner, i.e., each model in the sequence is only trained on those training samples that could not be certified robust by any of the previous models. The training algorithm is described by Wong et al. (2018) in appendix C (Algorithm 2) of their paper. We evaluate the efficacy of our attack algorithm (CasA) on cascade ensembles trained in a non-sequential manner. That is, each constituent model is trained independently on the entire train dataset, and each constituent only differs due to the randomness of initialization and of stochastic gradient descent during training. Besides this difference, the code, architecture, hyperparameters, and data used for training are the same as that used by Wong et al. (2018). For every combination of dataset, architecture, and $\epsilon$ value, we train three constituent models, and use them to construct non-sequentially trained cascading ensembles.

Table 4 shows the results of running our attack on such cascades with non-sequentially trained constituents. We make the following observations:

Table 4: Attack results on non-sequentially trained cascade ensembles.

| Dataset | Model | $\ell_p$ | $\epsilon$ | unsound CRA(%) | FPR(%) | Acc(%) | ERA(%) |
|---------|-------|----------|------------|----------------|--------|--------|--------|
| MNIST | Small,Exact | $\ell_\infty$ | 0.1 | 97.07 | 0.32 | 99.14 | 98.1 |
| MNIST | Small | $\ell_\infty$ | 0.1 | 96.25 | 0.27 | 98.69 | 97.4 |
| MNIST | Small | $\ell_\infty$ | 0.3 | 66.09 | 3.60 | 84.77 | 72.97 |
| CIFAR10 | Small | $\ell_\infty$ | 2/255 | 55.53 | 6.43 | 62.05 | 54.59 |
| CIFAR10 | Small | $\ell_\infty$ | 8/255 | 25.3 | 9.05 | 27.44 | 23.35 |
| MNIST | Small,Exact | $\ell_2$ | 1.58 | 47.42 | 0.0002 | 87.78 | 73.56 |
| MNIST | Small | $\ell_2$ | 1.58 | 47.4 | 0.0 | 87.5 | 72.93 |
| CIFAR10 | Small | $\ell_2$ | 36/255 | 52.33 | 3.29 | 55.88 | 53.71 |

- The non-sequentially trained ensembles continue to be unsound and our attack is able to find adversarial inputs (demonstrated by non-zero FPR).
- The success rate of our attack is much lower than on the sequentially-trained models shared by Wong et al. (2018).
- The unsound CRA of these ensembles is comparable to that of the sequentially-trained models.

We hypothesize that our attack demonstrates much higher success rates on sequentially-trained models because, when trained sequentially, it is likely that the later models in the cascade are degenerate, i.e., are very robust but with low accuracy (similar to constant functions). Then, to attack the ensemble, we just need to find an attack input where the initial models cannot be certified, since the remaining degenerate models are typically robust and inaccurate.

The degeneracy of the later models in the sequentially-trained ensemble may also explain why the unsound CRAs of the two kinds of ensembles are comparable. Models are trained in a sequential manner to enhance their "diversity". However, due to the degeneracy of the later models, the sequentially-trained ensembles likely end up being about only as diverse as the non-sequentially trained ensembles.

Finally, we note that the sequential style of training cascade ensembles is quite natural. In fact, both, Wong et al. (2018) and Blum et al. (2022) train models in a sequential manner. But these results suggest that sequentially training may make it easier to exploit the unsoundness of cascading ensembles.

## D  WEIGHTED VOTING ENSEMBLE: LEARNING WEIGHTS

The weights $w$ in $\mathcal{E}_V^w$ determines the importance of each constituent classifier in the ensemble. Given a set of $k$ labeled inputs, $S_k$ (e.g. the training set), we would like to learn the optimal weights $w$ that maximize the ensembler's CRA (Def. 2.2) over $S_k$. When $S_k$ resembles the true distribution of the test points, the learned $w$ is expected to be close to the optimal weights that maximizes the CRA of the test set. Weight optimization over $S_k$ naturally takes the following form.

$$\max_{w \in [0,1]^N} \frac{1}{k} \sum_{(x_i, y_i) \in S_k} \mathbb{1}\left[ \mathcal{E}_V^w(\tilde{F}^{(0)}, ..., \tilde{F}^{(N-1)})(x_i) = (y_i, 1) \right] \tag{15}$$

For the indicator to output 1, it is required that the margin of votes be greater than 0, i.e. $\Delta_{x_i}^w(y_i) := v_{x_i}^w(y_i, 1) - v_{x_i}^w(*, 0) - \max_{j \neq y_i}\{v_{x_i}^w(j, 1)\} > 0$. Namely, the votes for the class $y_i$, i.e. $v_{x_i}^w(y_i, 1)$, must be greater than the votes for all other classes i.e. $\max_{j \neq y_i}\{v_{x_i}^w(j, 1)\}$ plus the votes for non-robust predictions $v_{x_i}^w(*, 0)$ as discussed in Def. 5.1. Eq.(15) then becomes:

$$\max_{w \in [0,1]^N} \frac{1}{k} \sum_{(x_i, y_i) \in S_k} \mathbb{1}\left[ \Delta_{x_i}^w(y_i) > 0 \right] \tag{16}$$

The indicator function is not differentiable so we replace it with a differentiable and monotonically increasing function $s$, which leads to Eq. 17.

$$\max_{w \in [0,1]^N} \frac{1}{k} \sum_{(x_i, y_i) \in S_k} s(\Delta_{x_i}^w(y_i)) \tag{17}$$

In this paper, we choose $s$ to be the sigmoid function $\sigma_t$ where $t$ is the temperature only for negative inputs, i.e., $\sigma_t(x) := \sigma(x)$ if $x > 0$ and $\sigma(x/t)$ otherwise, where $\sigma$ is the standard sigmoid function. Sigmoid is non-negative so margins with opposite signs do not cancel, and it also avoids biasing training towards producing larger margins on a small number of points. Indeed, vanishing gradients are

Table 5: Results on models pre-trained by Wong et al. (2018) for $\ell_\infty$ robustness.

| Dataset | Model | $\epsilon$ | Single Model | | Cascading | | | Uniform Voting | | Weighted Voting | |
|---------|-------|------------|--------------|--------|-----------|--------|--------|----------------|--------|-----------------|--------|
| | | | CRA(%) | Acc(%) | unsound CRA(%) | Acc(%) | ERA(%) | CRA(%) | Acc(%) | CRA(%) | Acc(%) |
| MNIST | Small, Exact | 0.1 | 95.54 | 98.96 | 96.33 | 96.62 | 11.17 | 0.01 | 61.68 | 95.54 | 61.68 |
| MNIST | Small | 0.1 | 94.94 | 98.79 | 96.07 | 96.24 | 17.51 | 9.29 | 65.85 | 94.94 | 98.79 |
| MNIST | Large | 0.1 | 95.55 | 98.81 | 96.27 | 96.42 | 13.27 | 10.12 | 63.89 | 95.55 | 98.81 |
| MNIST | Small | 0.3 | 56.21 | 85.23 | 65.41 | 65.80 | 7.67 | 11.48 | 56.46 | 56.21 | 85.23 |
| MNIST | Large | 0.3 | 58.02 | 88.84 | 65.50 | 65.50 | 9.65 | 26.95 | 65.97 | 58.02 | 88.84 |
| CIFAR10 | Small | 2/255 | 46.43 | 60.86 | 56.65 | 56.65 | 50.13 | 18.58 | 40.88 | 46.43 | 60.86 |
| CIFAR10 | Large | 2/255 | 52.65 | 67.70 | 64.88 | 65.14 | 58.15 | 18.07 | 48.92 | 52.65 | 67.70 |
| CIFAR10 | Small | 8/255 | 20.58 | 27.60 | 28.32 | 28.32 | 23.79 | 10.78 | 24.11 | 19.00 | 23.78 |
| CIFAR10 | Large | 8/255 | 16.04 | 19.01 | 20.83 | 20.83 | 17.15 | 5.18 | 21.01 | 16.04 | 19.01 |

Table 6: Results on models pre-trained by Wong et al. (2018) for $\ell_2$ robustness.

| Dataset | Model | $\epsilon$ | Single Model | | Cascading | | | Uniform Voting | | Weighted Voting | |
|---------|-------|------------|--------------|--------|-----------|--------|--------|----------------|--------|-----------------|--------|
| | | | CRA(%) | Acc(%) | unsound CRA(%) | Acc(%) | ERA(%) | CRA(%) | Acc(%) | CRA(%) | Acc(%) |
| MNIST | Small, Exact | 1.58 | 43.52 | 88.14 | 75.58 | 80.43 | 43.46 | 6.42 | 74.25 | 43.52 | 88.14 |
| MNIST | Small | 1.58 | 43.34 | 87.73 | 74.66 | 79.07 | 45.73 | 6.74 | 74.24 | 43.34 | 87.73 |
| MNIST | Large | 1.58 | 43.96 | 88.39 | 74.50 | 74.99 | 35.81 | 6.35 | 65.99 | 43.96 | 88.39 |
| CIFAR10 | Small | 36/255 | 46.05 | 54.39 | 49.89 | 51.37 | 49.27 | 11.47 | 38.74 | 46.05 | 54.39 |
| CIFAR10 | Large | 36/255 | 50.26 | 60.14 | 58.72 | 58.76 | 57.17 | 10.56 | 39.74 | 50.26 | 60.14 |
| CIFAR10 | Resnet | 36/255 | 51.65 | 60.7 | 58.65 | 58.69 | 56.28 | 22.74 | 46.71 | 51.65 | 60.7 |

useful on points around large positive margins, so the temperature is only applied on negative inputs. This leads us to Eq. 18, the optimization objective we solve for optimal weights $w^*$.

$$w^* = \underset{w \in [0,1]^N}{\operatorname{argmax}} \frac{1}{k} \sum_{(x_i, y_i) \in S_k} \sigma_t(\Delta_{x_i}^w(y_i)) \tag{18}$$

## E  WEIGHTED VOTING ENSEMBLE: EMPIRICAL RESULTS

The goal of these experiments is to evaluate the efficacy of our sound voting ensemble. For our experiments, we use the pre-trained ensemble constituent models made available by Wong et al. (2018) to construct three kinds of ensembles, namely, cascading ensembles, uniform voting ensembles, and weighted voting ensembles. The weights for the weighted voting ensemble are learned in the manner described in Appendix D. We report certified robust accuracy (CRA) and standard accuracy (Acc) for each ensemble as well as for the best constituent model. Note that all these ensembles are query-access but only the uniform voting and weighted voting ensembles are soundness-preserving. Consequently, the CRA reported for the cascading ensemble grossly overestimates the actual CRA as demonstrated by our attack results. We always set the temperature to 1e5 and learning rate to 1e-2 when learning the weights as described in Appendix D.

Table 5 shows the results for $\ell_\infty$ robustness. Each row in the table represents a specific combination of dataset (MNIST or CIFAR-10), architecture (Small or Large convolutional networks or Resnet), and $\epsilon$ value used for local robustness certification. Table 6 shows the results for $\ell_2$ robustness using constituent models pre-trained by Wong et al. (2018).

**Summary of Results.** We see from Tables 5 and 6 that while the cascading ensemble appears to improve upon the CRA of the single best model in the ensemble, these numbers are misleading due to the unsoundness of the certification mechanism. The CRA for the uniform voting and weighted ensembles are consistently lower than that reported by the cascading ensemble, and in many cases, significantly so. Uniform voting ensembles stand-out for their low CRA but there is a simple explanation for these results. The constituent models are trained by Wong et al. (2018) in a cascading manner, i.e., later constituent models are trained on only those points that cannot be certified by the previous models. This strategy causes the subset of inputs labeled correct and certifiably robust by each constituent model to have minimal overlap. However, voting ensembles need these input subsets to strike the right balance between diversity and overlap for improving the CRA .

Another interesting observation is that, in most cases, the CRA of the weighted voting ensemble and the single best constituent model are the same. This is again a consequence of the cascaded manner in which the constituent models are trained. The first model in the cascade typically vastly outperforms

Table 7: Results on non-sequentially trained models for $\ell_\infty$ robustness.

| Dataset | Model | $\epsilon$ | Single Model | | Cascading | | | Uniform Voting | | Weighted Voting | |
| | | | CRA(%) | Acc(%) | unsound CRA(%) | Acc(%) | ERA(%) | CRA(%) | Acc(%) | CRA(%) | Acc(%) |
|---|---|---|---|---|---|---|---|---|---|---|---|
| MNIST | Small, Exact | 0.1 | 95.61 | 99.02 | 97.07 | 99.14 | 98.1 | 95.56 | 99.16 | 95.54 | 98.96 |
| MNIST | Small | 0.1 | 94.94 | 98.79 | 96.25 | 98.69 | 97.4 | 94.46 | 98.78 | 94.94 | 98.79 |
| MNIST | Small | 0.3 | 56.21 | 85.23 | 66.09 | 84.77 | 72.97 | 55.24 | 85.02 | 56.21 | 85.23 |
| CIFAR10 | Small | 2/255 | 46.43 | 60.86 | 55.49 | 62.06 | 54.59 | 43.48 | 62.79 | 42.35 | 61.09 |
| CIFAR10 | Small | 8/255 | 21.04 | 28.29 | 25.11 | 27.73 | 23.35 | 20.44 | 28.32 | 21.04 | 28.29 |

Table 8: Results on non-sequentially trained models for $\ell_2$ robustness.

| Dataset | Model | $\epsilon$ | Single Model | | Cascading | | | Uniform Voting | | Weighted Voting | |
| | | | CRA(%) | Acc(%) | unsound CRA(%) | Acc(%) | ERA(%) | CRA(%) | Acc(%) | CRA(%) | Acc(%) |
|---|---|---|---|---|---|---|---|---|---|---|---|
| MNIST | Small, Exact | 1.58 | 43.52 | 88.14 | 47.42 | 87.78 | 73.56 | 42.71 | 88.19 | 43.52 | 88.14 |
| MNIST | Small | 1.58 | 43.34 | 87.73 | 47.40 | 87.50 | 72.93 | 42.87 | 87.99 | 43.34 | 87.73 |
| CIFAR10 | Small | 36/255 | 46.05 | 54.39 | 52.33 | 55.88 | 53.71 | 37.07 | 57.40 | 42.12 | 54.65 |

the subsequent models. Moreover, as already mentioned, the constituent models have almost no overlap in the input regions where they perform well, and their presence only ends up harming the performance of the voting ensemble. As a consequence, the optimal normalized weights, learned by solving the optimization problem described in Appendix D, typically assign all the mass to the first model. The detailed weights for each of the weighted voting ensemble are given in Tables 9, 10, 11, and 12.

These results suggest two takeaway messages. First, the cascaded strategy of Wong et al. (2018) for training constituent models is in conflict with the requirement that constituent models overlap in their behavior for voting ensembles to be effective. This gives up hope that if the constituent models are suitably trained, voting ensembles can improve the CRA. We leave this exploration for future work. Second, even if the constituent models do not show the right balance between diversity and similarity, our weight learning procedure ensures that the performance of the weighted voting ensemble is no worse than the single best constituent model. Ideally, we would like the weights to be equally distributed since this conveys that every constituent in the ensemble has something to contribute. But, in the worst case, the weights play the role of a model selection procedure, assigning zero weights to constituent models that do not contribute to the ensemble.

**Non-Sequential Training.** We conduct another set of experiments where instead of using the constituent models pre-trained by Wong et al. (2018), we train them ourselves in a non-sequential manner. That is, each constituent model is trained on the entire train dataset, and each constituent only differs due to the randomness of initialization and of stochastic gradient descent during training. Besides this difference, the code, architecture, hyperparameters, and data used for training are the same as that used by Wong et al. (2018). For every combination of dataset, architecture, and $\epsilon$ value, we train three constituent models, and use them to construct cascading, uniform voting, and weighted voting ensembles.

Table 7 shows the results for $\ell_\infty$ robustness using non-sequentially trained constituent model ands Table 8 shows the results for $\ell_2$ robustness. We observe that, for non-sequentially trained models, the CRA of uniform voting and weighted voting ensembles are comparable, and similar to the CRA of the single best constituent model in the ensemble. In this case, the constituent models have too much overlap and almost no diversity. These results reaffirm our observation that voting ensembles require a balance between diversity and similarity to be effective.

## F   AN ALTERNATE FORMULATION OF UNIFORM VOTING ENSEMBLER

**Definition F.1** (Permutation-based Cascading Ensembler). Let $\tilde{F}^{(0)}$, $\tilde{F}^{(1)}$, ... , $\tilde{F}^{(N-1)}$ be $N$ certifiable classifiers and $N$ is odd. Suppose $\Pi$ is the set of all permutations of $\{0,1,...,N-1\}$. A permutation-based cascading ensemble $\mathcal{E}_P : \tilde{\mathbb{F}}^N \to \tilde{\mathbb{F}}$ is defined as follows

$$\mathcal{E}_P(\tilde{F}^{(0)},\tilde{F}^{(1)},...,\tilde{F}^{(N-1)})(x) := \begin{cases} \tilde{F}^{(\pi_0)}(x) & \text{if } \exists \pi \in \Pi. \; \mathtt{c2}(\pi)=1 \\ (\tilde{F}^{(\pi_0)}_{\mathsf{label}}(x),0) & \text{if } \nexists \pi' \in \Pi. \; \mathtt{c2}(\pi')=1 \wedge \exists \pi \in \Pi. \; \mathtt{c1}(\pi)=1 \\ (*,0) & \text{otherwise} \end{cases}$$

Table 9: Learned weights for weighted voting ensemble with models pre-trained by Wong et al. (2018) for $\ell_\infty$ robustness.

| Dataset | Model | $\epsilon$ | number of models | weights |
|---|---|---|---|---|
| MNIST | Small, Exact | 0.1 | 6 | [0.996, 0.003, 0.000, 0.001, 0.000, 0.000] |
| MNIST | Small | 0.1 | 7 | [0.996, 0.000, 0.000, 0.003, 0.000, 0.000, 0.001] |
| MNIST | Large | 0.1 | 5 | [0.996, 0.002, 0.001, 0.000, 0.001] |
| MNIST | Small | 0.3 | 3 | [0.995, 0.003, 0.002] |
| MNIST | Large | 0.3 | 3 | [0.948, 0.008, 0.044] |
| CIFAR10 | Small | 2/255 | 5 | [0.995, 0.002, 0.001, 0.001, 0.001] |
| CIFAR10 | Large | 2/255 | 4 | [0.994, 0.003, 0.001, 0.002] |
| CIFAR10 | Small | 8/255 | 3 | [0.003, 0.995, 0.002] |
| CIFAR10 | Large | 8/255 | 3 | [0.995, 0.002, 0.003] |

Table 10: Learned weights for weighted voting ensembles with models pre-trained by Wong et al. (2018) for $\ell_2$ robustness.

| Dataset | Model | $\epsilon$ | number of models | weights |
|---|---|---|---|---|
| MNIST | Small, Exact | 1.58 | 6 | [0.995, 0.001, 0.001, 0.000, 0.003, 0.001] |
| MNIST | Small | 1.58 | 6 | [0.995, 0.001, 0.002, 0.001, 0.001, 0.000] |
| MNIST | Large | 1.58 | 6 | [0.996, 0.000, 0.001, 0.002, 0.000, 0.001] |
| CIFAR10 | Small | 36/255 | 2 | [0.994, 0.006] |
| CIFAR10 | Large | 36/255 | 6 | [0.994, 0.002, 0.001, 0.001, 0.001, 0.001] |
| CIFAR10 | Resnet | 36/255 | 4 | [0.994, 0.004, 0.001, 0.001] |

Table 11: Learned weights for weighted voting ensemble with non-sequentially trained models for $\ell_\infty$ robustness.

| Dataset | Model | $\epsilon$ | number of models | weights |
|---|---|---|---|---|
| MNIST | Small, Exact | 0.1 | 3 | [0.710, 0.131, 0.159] |
| MNIST | Small | 0.1 | 3 | [0.694, 0.154, 0.152] |
| MNIST | Small | 0.3 | 3 | [0.908, 0.061, 0.031] |
| CIFAR10 | Small | 2/255 | 3 | [0.011, 0.956, 0.0323] |
| CIFAR10 | Small | 8/255 | 3 | [0.042, 0.087, 0.871] |

Table 12: Learned weights for weighted voting ensembles with non-sequentially trained models for $\ell_2$ robustness.

| Dataset | Model | $\epsilon$ | number of models | weights |
|---|---|---|---|---|
| MNIST | Small, Exact | 1.58 | 3 | [0.695, 0.159, 0.146] |
| MNIST | Small | 1.58 | 3 | [0.660, 0.196, 0.144] |
| CIFAR10 | Small | 36/255 | 3 | [0.003, 0.002, 0.995] |

where $*$ is a random label selected from $\mathcal{Y}$[7] , $\pi_0$ refers to the first element of the permutation $\pi$,

$$\texttt{c1}(\pi) := \begin{cases} 1 & \text{if } \exists j. \, (\frac{N+1}{2} \leq j \leq N-1) \wedge (\forall i < j. \, \tilde{F}_{\text{label}}^{(\pi_i)}(x) = \tilde{F}_{\text{label}}^{(\pi_j)}(x)) \\ 0 & \text{otherwise} \end{cases} \tag{19}$$

$$\texttt{c2}(\pi) := \begin{cases} 1 & \text{if } \exists j. \, (\frac{N+1}{2} \leq j \leq N-1) \wedge (\forall i < j. \, \tilde{F}^{(\pi_i)}(x) = \tilde{F}^{(\pi_j)}(x)) \wedge (\tilde{F}_{\text{cert}}^{(\pi_j)}(x) = 1) \\ 0 & \text{otherwise} \end{cases} \tag{20}$$

and $i, j \in \{0, 1, ..., N-1\}$.

**Theorem F.2.** *The permutation-based cascading ensembler $\mathcal{E}_P$ is a soundness-preserving ensembler.*

*Proof.* Let $\tilde{F} := \mathcal{E}_P(\tilde{F}^{(0)}, \tilde{F}^{(1)}, ..., \tilde{F}^{(N-1)})$. For $\tilde{F}$ we want to show that,

$$\forall x \in \mathbb{R}^d, \tilde{F}_{\text{cert}}(x) = 1 \Longrightarrow \tilde{F}_{\text{label}} \text{ is } \epsilon\text{-locally robust at } x. \tag{21}$$

W.L.O.G suppose $\tilde{F}_{\text{label}}(x) = y$. If $\tilde{F}(x) = (y, 1)$, let us assume that $\pi$ is the permutation such that $\texttt{c2}(\pi) = 1$. Let $k$ be the integer that makes $\texttt{c2}(\pi) = 1$ to be true. Thus

$$\tilde{F}(x) = (y, 1) \Longrightarrow (\frac{N+1}{2} \leq k \leq N-1) \wedge (\forall i < k, \tilde{F}^{(\pi_i)}(x) = \tilde{F}^{(\pi_k)}(x) = (y, 1)) \tag{22}$$

By our assumptions that $\tilde{F}^{(0)}, \tilde{F}^{(1)}, ..., \tilde{F}^{(N-1)}$ are sound, which are invariant to the permutation of these models. Therefore, by Def. 2.3, $\forall i \leq k$,

$$\tilde{F}^{(\pi_i)}(x) = (y, 1) \Longrightarrow (\forall x'. \, ||x' - x|| \leq \epsilon \Longrightarrow \tilde{F}_{\text{label}}^{(\pi_i)}(x') = \tilde{F}_{\text{label}}^{(\pi_i)}(x) = y) \tag{23}$$

Eq. (23) implies that $\forall x'$ s.t. $||x' - x|| \leq \epsilon$, the following statement is true

$$(\frac{N+1}{2} \leq k \leq N-1) \wedge (\forall i \leq k, \tilde{F}_{\text{label}}^{(\pi_i)}(x) = \tilde{F}_{\text{label}}^{(\pi_k)}(x') = y) \tag{24}$$

Plug the condition (24) into Def. F.1, we find that $\texttt{c1}(\pi) = 1$ for $x'$. Moreover, there cannot be a permutation $\pi'$ such that $\texttt{c1}(\pi') = \texttt{c2}(\pi') = 1 \wedge \tilde{F}_{\text{label}}^{(\pi_0')} \neq y$ since $k \geq \frac{N+1}{2}$. Therefore, $\tilde{F}_{\text{label}}(x') = \tilde{F}_{\text{label}}^{(\pi_0)}(x') = y$, and we arrive at the following statement,

$$\forall x'. \, ||x' - x|| \leq \epsilon \Longrightarrow \tilde{F}_{\text{label}}(x') = \tilde{F}_{\text{label}}(x) = y \tag{25}$$

which completes the proof for the soundness of $\tilde{F}_{\text{label}}$ at any $x$.

$\square$

---

[7]One can also return the plurality prediction of all models for the consideration of clean accuracy but the choice of $*$ will not change the relevant theorems.

