# OpenReview forum: "On the Perils of Cascading Robust Classifiers"
_ICLR.cc/2023/Conference — ICLR 2023 poster_

### Official Review · Reviewer_YgY8 · 2022-10-21

**Confidence:** 2
**Correctness:** 3
**Technical Novelty And Significance:** 2
**Empirical Novelty And Significance:** 3
**Recommendation:** 6

**Clarity, Quality, Novelty And Reproducibility:**

The paper seems reproducible, clear, and easy to read. It does point out some issue with the robust certification of cascade classifiers that is not previously known.

The result seems reproducible.



**Strength And Weaknesses:**

Strength
- The paper is straight forward, rigorous, and easy to follow.

Weaknesses
- I am not confident how important/widespread cascade robust classifiers are. If it is barely used, it may limit the scope of this paper.
- In Table 4 and 5, is the CRA for cascading ensemble the correct CRA or the overestimated CRA? It seems it is the latter one. If that's the case, I think it is necessary to compare the ERA with the CRA of weighted voting ensemble so that we can understand how effective is the weighted voting ensemble.
- It also seems that weighted voting have CRA and Acc less than or equal to a single model. So what could be the use case of this ensemble over a single model.

**Summary Of The Paper:**

This paper presents a counterexample to prove that certified cascading robust classifiers does not hold up to its certified robustness claim. Then, they proposed an attack method to show that on real data, models certified for cascading robust classifiers failed to held up to their certification. In other words, there are adversarial examples that makes the certified robust accuracy significantly lower than the certification. Finally, they proposed an alternative ensemble method, weighted voting ensemble, that is provably sound (can correctly provide a certified robust radius).

**Summary Of The Review:**

I think this is a well-written paper. However, the paper seems to have a limited scope and the proposed method seems not so useful.

---

> ### Author Response · Authors · 2022-11-09
> **Response to Reviewer YgY8**
>
> We thank the reviewer for their comments and feedback.
>
> > I am not confident how important/widespread cascade robust classifiers are. If it is barely used, it may limit the scope of this paper.
>
>
>
> [1] is a well-cited paper (372 citations as of today per Google Scholar) while [2] was recently published at NeurIPS 2022. We think this shows that the idea of cascaded ensembles for improving robustness is alive in the community that works on robust and trustworthy machine learning. Our work can help prevent the further propagation of this unsound approach and motivate the investigation of new sound black-box ensembling approaches.
>
>
>
> > In Table 4 and 5, is the CRA for cascading ensemble the correct CRA or the overestimated CRA? It seems it is the latter one. If that's the case, I think it is necessary to compare the ERA with the CRA of weighted voting ensemble so that we can understand how effective is the weighted voting ensemble.
>
>
>
> Tables 4 and 5 indeed report the overestimated CRA of the cascading ensemble. We will copy the ERA column from Table 1 to Table 4 and 5 so that direct comparison between ERA of the cascading ensemble and CRA of the weighted voting ensemble can be made.
>
> Note that the CRA of the weighted voting ensemble is in many cases higher than the ERA of the cascading ensemble. Since CRA represents a lower bound on the robust accuracy while ERA represents an upper bound, in such cases, the weighted voting ensemble conclusively outperforms the cascading ensemble. On the other hand, in the cases where CRA of weighted voting ensemble is lower than the ERA of the cascading ensemble, it is not possible to definitively conclude that cascading ensembles outperform weighted voting ensembles since all we know is that a lower bound on robust accuracy (CRA) is less than an upper bound on robust accuracy (ERA).
>
>
>
> > It also seems that weighted voting have CRA and Acc less than or equal to a single model. So what could be the use case of this ensemble over a single model.
>
>
>
> As we note in our response to reviewer *VBmo*, we present the weighted voting ensemble primarily as an alternative baseline black-box ensembling mechanism that is provably sound. Getting weighted voting ensembles to outperform single models requires the models in the ensemble to suitably balance diversity and similarity. While Example 5.4 shows that it is possible for a weighted ensemble, with a suitable balance of diversity and similarity, to vastly outperform single models, the algorithm for learning models that demonstrate this balance remains an open question for future research. We note that a weighted voting ensemble can always place all its weight on the single best model in the ensemble, and therefore, it is always possible to match the performance of the single best model in the ensemble.
>
>
>
> Also note that the primary contribution of this work is to point out the problems with cascading ensembles so that the community stops using them as a solution for improving robustness of models. The material presented on weighted voting ensemble is only to point out a potentially fruitful direction for future research on black-box ensembles that improve robustness.

---

> > ### Comment · Reviewer_YgY8 · 2022-11-23
> > **Thank you for the response**
> >
> > I would like to thank the authors for their responses. This paper pointed out solid flaws in cascade classifiers, and I am convinced it is important to fix these flaws. However, I think the proposed solution is not complete enough. In the best case, weighted voting may help with the problem (as in Example 5.4), but there is no detailed look into how to improve the balance between diversity and similarity. Due to the importance of having a better bound of the robustness of cascade ensembles, I would like to increase my score to 6.

---

### Official Review · Reviewer_VBmo · 2022-10-24

**Confidence:** 4
**Correctness:** 4
**Technical Novelty And Significance:** 3
**Empirical Novelty And Significance:** 3
**Recommendation:** 8

**Clarity, Quality, Novelty And Reproducibility:**

- The presentation is clear (maybe except for the "Surrogate Objectives" paragraph) and easy to follow.
- The issue with cascading ensembles has not been known previously as far as I am aware, thus the main result is novel.
- The used models are clearly referenced, and the methods are described well enough that manual re-implementation should not be an issue.
- The submitted code appears to be well documented and to contain all relevant experiments (though I didn't run it myself yet).

**Strength And Weaknesses:**

Strengths:

- The theoretical introduction and methodology are clearly presented and appear to be sound.

- The experimental evaluations are very strong, as (contrary to other counterexamples against similar certifications) they work with fixed input points and fixed radii. I.e. not finding specific, potentially non-sensical input images but evaluatin the actual test set.

Weaknesses/Issues/Suggestions/Questions:

- The paper is mostly about showing that a previously presented improvement of adversarial robustness certification is invalid.
I think it is very important to point out crucial issues with previously published and well cited papers, so I think that the submission is a good contribution for ICLR.
However, I am not sure how popular/relevant the cascading methods in [1] and [2] are currently and if they are nominally relevant when applied to ensembles of and compared with state of the art certifyable robusntess techniques. A bit more evidence for them being used would underline the importance of the submission.

- A more concrete overview of the attacked ensemble from [1] would be helpful. Particularly on how the sequence of models has been trained.
	- I would assume that if the models were trained independently (e.g. just from different seeds), the attack success rate on the cascading ensemble would be much smaller. Is that correct?

- It would be helpful to emphasize that cascade ensembling over different certification methods for one classifier, as e.g. done in [3], is sound.

- "We solve the optimization problems on lines 6 and 8 using projected gradient descent (PGD)" -- Why do you not use a stronger adversarial attack scheme, like AutoAttack [0]? Intuitively, the same considerations as for adversarials should apply for optimizing the objective for $\tilde{F}(x+\delta)$, right? Since and attack and not a defense is presented, this is not a major issue since the claims made by the paper could only be improved, but it could lead to improvements and might be the correct thing to do.

- The paragraph on "Surrogate Objectives" is very confusing to me and should potentially be rewritten.
	 - Optimizing the indicator functions does not seem to make sense: Would $argmax_\delta \mathbb{1}[\tilde{F}_{cert}^{(i)}(x+\delta)]$ not be realized for too many points, often including $\delta = \vec{0}$ since the function value is either $0$ or $1$?
	 - Before, completely black box models were considered, but now access to the logits is assumed; this new assumption should be stated.
	 - For the sake of demonstrating wrong certifications, it might make sense to use a specific objective for each model that appears in the counterexample.

- There are very small differences between the evaluations in Table 1 and the original numbers from [1]. How can they be explained? Potentially, the explanation should be included in the paper.

- The proposed weighted does not seem to be a convincing contribution.
	 - Its results are not better than using the single first model (correct?).
	 - Its construction and its theoretical soundness are not particularly surprising or insightful.
	 - The case presented in Example 5.4. shows that there are situations where a voting ensemble does better than a single constituent model. However it doesn't seem to be possible to decide if this is the case in a given situation (without knowing the test labels). There are certainly also many situations where it is the other way around, and the voting ensemble is worse than a single model.
	 - Still, it is a baseline alternative to cascading ensembling, and thus it is interesting to have it discussed and evaluated in this paper, I would just not see it as a recommendable alternative to using a single model, as I understand the results.

[0] Croce and Hein, ICML 2020: "Reliable evaluation of adversarial robustness with an ensemble of diverse parameter-free attacks"

[1] Wong et al., NeurIPS 2018: "Scaling provable adversarial defenses"

[2] Blum et al. 2022: "Boosting barely robust learners: A new perspective on adversarial robustness."

[3] Gowal et al. 2018: "On the effectiveness of interval bound propagation for training verifiably robust models."

**Summary Of The Paper:**

Cascading ensembling of certifiably adversarially robust classifiers, which has been used by previous works, is shown to be unsound, which is demonstrated by theoretical proof as as well as experimental results on real test set data.

**Summary Of The Review:**

- The paper points out an important mathematical issue with the method of cascading ensembling of certifiably adversarially robust classifiers and shows with convincing experiments that this issue is relevant in practice.
- This makes it an interesting contribution in my opinion.
- The proposed weighted voting ensembling method is to my understanding not convincing by the discussed theoretical properties or experimental results, but it does not really hurt the paper and its efficacy is not overclaimed.

---

> ### Author Response · Authors · 2022-11-09
> **Response to Reviewer VBmo**
>
> We thank the reviewer for their comments and feedback.
>
> > I am not sure how popular/relevant the cascading methods in [1] and [2] are currently and if they are nominally relevant when applied to ensembles of and compared with state of the art certifiable robustness techniques. A bit more evidence for them being used would underline the importance of the submission.
>
>
>
> [1] is a well-cited paper (372 citations as of today per Google Scholar) while [2] was recently published at NeurIPS 2022. We think this shows that the idea of cascaded ensembles for improving robustness is alive in the community that works on robust and trustworthy machine learning. Our work can help prevent the further propagation of this unsound approach and motivate the investigation of new sound black-box ensembling approaches.
>
>
> > A more concrete overview of the attacked ensemble from [1] would be helpful. Particularly on how the sequence of models has been trained. I would assume that if the models were trained independently (e.g. just from different seeds), the attack success rate on the cascading ensemble would be much smaller. Is that correct?
>
>
>
> Note that we use the pre-trained cascading ensembles provided by Wong et al. (2018) for our evaluation. Each model in the sequence is only trained on those training samples that could not be certified robust by any of the previous models. Wong et al. (2018) describe their algorithm (Algorithm 2) in the appendix C of their paper ([https://arxiv.org/pdf/1805.12514.pdf](https://arxiv.org/pdf/1805.12514.pdf)). We will add a brief description of the same in the revised version of our paper (and will share this revised version next week).
>
>
> We do expect that for models trained independently (i.e., each model trained on the entire dataset using different seeds), the attack success rate on the resulting ensemble will be smaller. At the same time, we also expect that the unsound CRA reported by such an ensemble would also be smaller. We are working on evaluating this empirically and expect to share the results next week before the discussion period ends.
>
>
>
>
> > It would be helpful to emphasize that cascade ensembling over different certification methods for one classifier, as e.g. done in [3], is sound.
>
>
>
> We will add this clarification.
>
>
>
> > Why do you not use a stronger adversarial attack scheme, like AutoAttack [0]? Intuitively, the same considerations as for adversarials should apply for optimizing the objective for $\tilde{F}(x+\delta)$ , right? Since an attack and not a defense is presented, this is not a major issue since the claims made by the paper could only be improved, but it could lead to improvements and might be the correct thing to do.
>
>
>
> The reviewer is correct that we could have used a stronger attack like AutoAttack. However, since our attack already turned out to be strong enough to point out the significant issues with cascading ensembles and support our theoretical finding, we did not evaluate the effect of stronger attacks.
>
>
>
> > The paragraph on "Surrogate Objectives" is very confusing to me and should potentially be rewritten.
>
>
> The inuition behind the design of our surrogate objectives is that, given a point $x$, the distance to the decision boundary of the model  (and therefore the local robustness of the model) is proportional to the margin (i.e., the difference) between the top two logit values output by the model. Therefore, to find a point $x'$ where the model is certified robust, a surrogate objective is to find an $x'$ that maximizes the margin between the top two logits. Similarly, to find a point $x'$ where the model cannot be certified robust, a surrogate objective is find an $x'$ such that the margin between the logits is minimized.
>
> Please let us know if this clarification helps and we can incorporate it in our text.

---

> > ### Author Response · Authors · 2022-11-09
> > **Response to Reviewer VBmo (continued)**
> >
> > >Optimizing the indicator functions does not seem to make sense: Would $argmax_\delta\mathbb{1}[\tilde{F}_{cert}^{(i)}(x+\delta)]$ not be realized for too many points, often including $\delta=0$ since the function value is either 0 or 1?
> >
> >
> >
> > A successful attack input $x’$ is one where model $\tilde{F}^{(i)}$ is certified robust at $x’$ whereas all models earlier in the sequence are not certified robust at $x’$. Note that we only try to attack those models in the cascade that predict a label other than $y$ at original input $x$ (line 3 of Algorithm 3.1).
> >
> > Also note that $\mathbb{1}[\tilde{F}_{cert}^{(i)}(x+\delta)]$ takes value 1 when
> >
> >  $\tilde{F}_{cert}^{(i)}(x+\delta)=1$
> >
> > (implying that $\tilde{F}^{(i)}$ is certified robust at $x+\delta$)
> >
> > and 0 when $\tilde{F}_{cert}^{(i)}(x+\delta)=0$
> >
> > (implying that $\tilde{F}^{(i)}$ is not certified robust at $x+\delta$).
> >
> > Therefore, any $\delta$ that causes $\mathbb{1}[\tilde{F}_{cert}^{(i)}(x+\delta)]=1$ partially meets the criteria for a successful attack (we also need to ensure that all the earlier models in the sequence are not certified robust).
> >
> > In other words, $argmax_\delta\mathbb{1}[\tilde{F}_{cert}^{(i)}(x+\delta)]$ is only realized at points $x+\delta$ where $\tilde{F}^{(i)}$ indeed can be certified robust.
> >
> >
> >
> > We realize that the use of the indicator function is superfluous since $\tilde{F}_{cert}^{(i)}$ is already a function that only takes values 0 or 1.Therefore, we can write the optimization objective simply as
> >
> > $argmax_\delta\tilde{F}_{cert}^{(i)}(x+\delta)$. We will update this in the paper.
> >
> >
> >
> >
> >
> > >Before, completely black box models were considered, but now access to the logits is assumed; this new assumption should be stated.
> >
> >
> >
> > We will add this assumption to the text.
> >
> >
> >
> > >For the sake of demonstrating wrong certifications, it might make sense to use a specific objective for each model that appears in the counterexample.
> >
> >
> > We are not completely sure what the reviewer has in mind here and would appreciate if the reviewer could elaborate their comment.
> >
> > Given an input $x$ where the cascading ensemble is certified robust,  our attack considers each model $i$ that appears in the set $\texttt{idxs}$ defined on line 3 of Algorithm 3.1 as a potential candidate that could be used to construct a counterexample. For each candidate considered, as mentioned earlier, we want to find an input $x'$ such that the candidate model is certified robust at $x'$ while the earlier models in the sequence are not certified robust at $x'$. Therefore, we solve the same optimization problem for each candidate (line 8 in Algorithm 3.1) . Only if the candidate model is the last model in the sequence, we consider a slightly relaxed version of the optimization problem (line 6 in Algorithm 3.1); instead of requiring that the candidate model be certified robust at $x'$, we only require that it have a different label from $y$ at $x'$, but continue to require that all earlier models not be certified robust at $x'$.
> >
> >
> > > There are very small differences between the evaluations in Table 1 and the original numbers from [1]. How can they be explained? Potentially, the explanation should be included in the paper.
> >
> >
> >
> > These small differences were surprising to us as well since we use the code and the models from [1] (although our hardware and PyTorch versions are different). We suspect that there might be some small sources of non-determinism in the evaluation and certification code provided by [1]. We will investigate this further and add a discussion in our paper.
> >
> >
> >
> > > The proposed weighted voting does not seem to be a convincing contribution.
> >
> >
> >
> > As the reviewer notes, we present the weighted voting ensemble primarily as an alternative baseline black-box ensembling mechanism that is provably sound. Getting weighted voting ensembles to outperform single models requires the models in the ensemble to suitably balance diversity and similarity. While Example 5.4 shows that it is possible for a weighted ensemble with a suitable balance of diversity and similarity to vastly outperform single models, the algorithm for learning models that demonstrate this balance remains an open question for future research. We note that a weighted voting ensemble can always place all its weight on the single best model in the ensemble, and therefore, it is always possible to match the performance of the single best model in the ensemble.

---

> > ### Comment · Reviewer_VBmo · 2022-11-25
> > **Re: Responses**
> >
> > Thank you for the detailed responses.
> >
> > >> I am not sure how popular/relevant the cascading methods in [1] and [2] are currently and if they are nominally relevant when applied to ensembles of and compared with state of the art certifiable robustness techniques. A bit more evidence for them being used would underline the importance of the submission.
> > > [1] is a well-cited paper (372 citations as of today per Google Scholar) while [2] was recently published at NeurIPS 2022. We think this shows that the idea of cascaded ensembles for improving robustness is alive in the community that works on robust and trustworthy machine learning. Our work can help prevent the further propagation of this unsound approach and motivate the investigation of new sound black-box ensembling approaches.
> >
> > As I mentioned in the original review, I am aware that [1] is a well cited paper. However, this is mainly because of its important improvements for individual networks. None of the papers (besides [2]) that cite it that I am aware of even mention the cascades. This is the reason why it would be very beneficial to have a commented list of some works that do use them.
> > Even leaderboards for exactly the provable adversarial robustness problem at hand like https://github.com/AI-secure/Certified-Robustness-SoK-Oldver do not show the cascade numbers of [1] but only the single model numbers.
> >
> > >> [training of the ensemble]
> > Thanks for the updated Appendix C, it is helpful to understand the methodology in [1].
> >
> >
> >
> > >>It would be helpful to emphasize that cascade ensembling over different certification methods for one classifier, as e.g. done in [3], is sound.
> >
> > > We will add this clarification.
> >
> > Thanks.
> >
> > >> Why do you not use a stronger adversarial attack scheme, like AutoAttack [0]? Intuitively, the same considerations as for adversarials should apply for optimizing the objective for, right? Since an attack and not a defense is presented, this is not a major issue since the claims made by the paper could only be improved, but it could lead to improvements and might be the correct thing to do.
> >
> > > The reviewer is correct that we could have used a stronger attack like AutoAttack. However, since our attack already turned out to be strong enough to point out the significant issues with cascading ensembles and support our theoretical finding, we did not evaluate the effect of stronger attacks.
> >
> > I am still of the opinion that it would be better to use a more up to date attack scheme in order to provide results that are as close to the true robustness as possible.
> >
> > >> [paragraph on "Surrogate Objectives"]
> >
> > Thanks for the improvements in the paragraph.

---

> ### Author Response · Authors · 2022-11-09
> **Attack evaluation on independently trained models**
>
> The table below shows the results of running our attack on cascade ensembles with 3 models each, where each model was trained independently and on the **entire** training dataset.
>
>
> | Dataset | Model    | $\ell_p$,  $\epsilon$      | unsound CRA (%) | FPR (%) | Acc (%) | ERA (%) |
> |----------|----------|--------------|-----------------|---------|---------|---------|
> | MNIST | Small,Exact | $\ell_\infty$, 0.1    | 97.07           | 0.32    | 99.14   | 98.1    |
> | MNIST | Small       | $\ell_\infty$, 0.1    | 96.25           | 0.27    | 98.69   | 97.4    |
> | MNIST | Small       | $\ell_\infty$, 0.3    | 66.09           | 3.60    | 84.77   | 72.97   |
> | CIFAR10 | Small      | $\ell_\infty$, 2/255 | 55.53           | 6.43    | 62.05   | 54.59   |
> | CIFAR10 | Small    | $\ell_\infty$, 8/255  | 25.3            | 9.05    | 27.44   | 23.35   |
> | CIFAR10 | Small     | $\ell_2$, 36/255    | 52.33           | 3.29    | 55.88   | 53.71   |
>
>
>
> We make the following observations:
> 1. The non-sequentially trained ensembles continue to be unsound and our attack is able to find adversarial inputs (demonstrated by non-zero FPR).
> 2. The success rate of our attack is much lower than on the sequentially-trained models shared by [1].
> 3. The unsound CRA of these ensembles is comparable to that of the sequentially-trained models.
>
> We hypothesize that our attack demonstrates much higher success rates on sequentially-trained models because, when trained sequentially, it is likely that the later models in the cascade are degenerate, i.e., are very robust but with low accuracy (similar to constant functions). Then, to attack the ensemble, we just need to find an attack input $x'$ where the initial models cannot be certified, since the remaining degenerate models are typically robust and inaccurate.
>
> The degeneracy of the later models in the sequentially-trained ensemble may also explain why the unsound CRAs of the two kinds of ensembles are comparable. Models are trained in a sequential manner to enhance their "diversity". However, due to the degeneracy of the later models, the sequentially-trained ensembles likely end up being about only as diverse as the non-sequentially trained ensembles.
>
> Finally, we note that the sequential style of training cascade ensembles is quite natural. In fact, both, [1] and [2] train models in a sequential manner. But these results suggest that sequentially training may make it easier to exploit the unsoundness of cascading ensembles.

---

> > ### Comment · Reviewer_VBmo · 2022-11-25
> > **Re: Attack evaluation on independently trained models**
> >
> > Thanks for the evaluation on the independently trained standard models and the preciding more detailed explanation of the original method in Appendix C. The success rate being much lower than in the original case is expected, but the numbers and discussion in the above answer are helpful for understanding of the method. That those cascade certifications are still unsound is also an interesting observation.

---

### Official Review · Reviewer_XiH9 · 2022-10-24

**Confidence:** 3
**Correctness:** 3
**Technical Novelty And Significance:** 3
**Empirical Novelty And Significance:** 4
**Recommendation:** 6

**Clarity, Quality, Novelty And Reproducibility:**

The clarity, quality, novelty and reproducibility are all good. Overall, this is an interesting paper.

**Strength And Weaknesses:**

Strengthes:
- The layout of this paper is good and easy to follow, pleasant to read and the idea is interesting. In fact, I am surprised such mistake in cascading classifiers are pointed out until now.
- The completeness of this paper is good: the authors first provided a theoretically sound proof; followed by a practical algorithm to find the adversarial examples; and finally intensive experiments are made to show that such algorithm is indeed able to


Weaknesses:
- Does (Blum et al. 2022) also employ the erroneous cascading ensemble? I roughly checked their paper and it seems not the case.
- It looks like even in (Wong 2018), the cascading ensemble was still claimed as one of the smaller contributions (only mentioned in one paragraph). So I think the negative impact of the unsoundness is limited ==> not sure whether publishing in a workshop is a better option (but I left this for authors and ACs to decide).
- Another question is -- instead of defining $\tilde{F}_{\text{cert}}=1$

as  $\tilde{F}_{label}$ to be epsilon-locally robust, can we define

$\tilde{F}_{cert}=1$

to be
$\tilde{F}_{label}\text{ is } \epsilon$-locally robust AND

$\tilde{F}_{label}$ equals to the ground truth $y$?

Because if the classifier makes a mistake, then we don’t need to care about its local stability at all. By strengthening the definition, the cascading classifier seems to be correct again.

**Summary Of The Paper:**

This paper pointed out a fallacy that cascading robust classifier is provably robust to epsilon perturbation. The authors first pointed out a counter example of such method and then provided a systematical algorithm to generate the adversarial examples from a sequence of provably sound classifiers. This paper essentially showed that chaining the epsilon-robust classifiers are by itself not epsilon-robust.

**Summary Of The Review:**

There aren't a lot to comment for this submission, after all this is a correction of previous misbelief.

---

> ### Author Response · Authors · 2022-11-09
> **Response to Reviewer XiH9**
>
> We thank the reviewer for their comments and feedback.
>
> > Does (Blum et al. 2022) also employ the erroneous cascading ensemble?
>
>
>
> Blum et al. (2022) indeed employ the erroneous cascading ensemble. Consider the definition of their cascade predictor CAS in Algorithm 1 ([https://arxiv.org/pdf/2202.05920.pdf](https://arxiv.org/pdf/2202.05920.pdf)). Just like Wong et al. (2018), they construct a cascade of selective classifiers where each selective classifier only makes a prediction at point $x$ if it can be certified locally robust at $x$. The only difference is that Blum et al. (2022) use a more general definition of local robustness, going beyond $p$-norm bounded robustness. Accordingly, instead of checking whether the classifier is $\ell_2$ or $\ell_\infty$ locally robust at $x$, they check if the classifier is robust in a region defined by $\mathcal{U}^{-1}(x)$. But the problems that we highlight with cascading robust classifiers are not restricted to $p$-norm bounded notions of robustness.
>
>
>
> > In (Wong 2018), the cascading ensemble was still claimed as one of the smaller contributions (only mentioned in one paragraph)
>
>
>
> The main tables in (Wong et al. 2018) (Table 2 and Table 4) as well as one of the main graphs (Figure 3) all report results for cascading ensembles, and their best reported numbers for robust error are contingent on the use of cascading ensembles.
> Further, this paper is well-cited (372 citations as of today per Google Scholar), so unless the issue with cascading ensembles is pointed out to the community at a major conference like ICLR, researchers and engineers are likely to continue adopting this unsound idea. As a case in point, (Blum et al. 2022), published in NeurIPS 2022, also builds on the idea of cascading robust classifiers. Our work will hopefully prevent the future use of cascading ensembles for improving robustness.
>
>
> > Instead of defining $\tilde{F}_{cert}=1$
>
> > as $\tilde{F}_{label}$ to be $\epsilon$-locally robust,
>
> > can we define $\tilde{F}_{cert}=1$ to be
>
> > $\tilde{F}_{label}$ is $\epsilon$-locally robust AND
>
> > $\tilde{F}_{label}$ equals to the ground truth $y$?
>
> While the definition of $\tilde{F}_{cert}$ proposed by the reviewer is reasonable, this definition is not operational because we do not know the ground truth at inference time. Moreover, our definition is also the one used by Wong et al. (2018) and Blum et al. (2022) in the design of their cascading ensembles.

---

### Author Response · Authors · 2022-11-18
**Uploaded revised version of the paper**

Dear reviewers,

Thank you for your comments that have helped improve this paper. We have uploaded a revised version of the paper with the changes promised in our response.

We made the following changes:
1. Added a footnote in Sec 2.2 saying that a cascade of certification techniques applied to a single model is sound.
2. Added a comment under **Surrogate Objectives** in Sec 3 saying that we need to assume access to logit scores for our attack.
3. Removed the superfluous indicator function from the surrogate objectives in Sec 3.
4. Added a note in Sec 4 under **Summary of Results** saying that our results show small differences compared to the numbers in Wong et al. (2018).
5. Added Appendix C briefly describing how cascading ensembles are trained in a sequential manner + describing attack results on cascades with non-sequentially and independently trained constituent models.
6. Added ERA column in tables 5,6,7 in the appendix.

---

### Decision · Program_Chairs · 2023-01-20

**Decision:**

Accept: poster

**Justification For Why Not Higher Score:**

* The topic of the paper is arguably "niche", so an oral may not be of interest for the wider ICLR community.
*  The cascading method's significance that the paper focuses on is arguably limited.

**Justification For Why Not Lower Score:**

* Original and surprising observation.
* Solid theoretical justifications.
* Solid empirical validation.
* Good support from all reviewers.
* Manuscript of good quality.

**Metareview: Summary, Strengths And Weaknesses:**

The reviewers and meta reviewer all appreciated the quality of the work with a clear, well-written manuscript, focusing on an original and surprising observation, namely that cascading ensembles of certifiably adversarially robust classifiers is not sound
Moreover, the method is supported by both solid theoretical justifications and experimental validations.

They thank the authors for their response and their efforts during the rebuttal phase, which was helpful to improve the submission and clarify concerns (e.g., more background about previous work & cascading ensembles, discussion about significance and new results for independently trained models). The reviewers and meta reviewer unanimously recommend the paper for acceptance.

As suggested by some of the reviewers, it would be important to improve the paper by underlining the cascading method's significance which seems not to have been broadly used in the past few years.

**Note From Pc:**

if the above contains the word "oral" or "spotlight" please see: "oral" presentation means -> notable-top-5% and "spotlight" means -> notable-top-25%. As stated in our emails, we are disassociating presentation type from AC recommendations